# A Textbook Remedy for Domain Shifts: Knowledge Priors for Medical Image Analysis

**Yue Yang, Mona Gandhi, Yufei Wang, Yifan Wu, Michael S. Yao,**
**Chris Callison-Burch, James C. Gee, Mark Yatskar**

University of Pennsylvania
`yueyang1996.github.io/knobo`

## Abstract

While deep networks have achieved broad success in analyzing natural images, when applied to medical scans, they often fail in unexpected situations. We investigate this challenge and focus on model sensitivity to domain shifts, such as data sampled from different hospitals or data confounded by demographic variables such as sex, race, etc, in the context of chest X-rays and skin lesion images. A key finding we show empirically is that existing visual backbones lack an appropriate prior from the architecture for reliable generalization in these settings. Taking inspiration from medical training, we propose giving deep networks a prior grounded in explicit medical knowledge communicated in natural language. To this end, we introduce **Kno**wledge-enhanced **Bo**ttlenecks (**KnoBo**), a class of concept bottleneck models that incorporates knowledge priors that constrain it to reason with clinically relevant factors found in medical textbooks or PubMed. KnoBo uses retrieval-augmented language models to design an appropriate concept space and an automatic training procedure for recognizing the concept. We evaluate different resources of knowledge and recognition architectures on a broad range of domain shifts across 20 datasets. In our comprehensive evaluation with two imaging modalities, KnoBo outperforms fine-tuned models on confounded datasets by 32.4 % on average. Finally, evaluations reveal that PubMed is a promising resource for making medical models less sensitive to domain shift, outperforming other resources on both diversity of information and final prediction performance.

## 1 Introduction

Robustness to domain shifts is a key property for models operating on medical images because transfer scenarios arise widely. Deep networks have achieved broad success in analyzing natural images (everyday human contexts), but when applied to medical scans, they often substantially degrade under distribution shift [86, 15]. Medical datasets are small, and unidentified confounds in them combined with model misspecification can dramatically degrade performance [46, 21, 24]. Such failure erodes confidence as models do not learn the right information from training data, hampering adoption by medical professionals. We study such problems by investigating the performance of systems in the presence of confounded data and address a main shortcoming we discover.

Model sensitivity to domain shift can be measured by introducing synthetic confounds into data and evaluating on samples where the confound misleads the model. For example, in Figure 1, we introduce confounded datasets for chest X-ray and skin lesion images where, during training, positive data is sampled from one group and negative from another. This association is reversed at testing time, creating an adversarial out-of-distribution (OOD) evaluation. In 5 such constructed confounds per modality, covering scenarios of race, sex, age, scan position, and hospital, we find models unable to generalize well, dropping over 63% on average over an in-distribution (ID) evaluation.

38th Conference on Neural Information Processing Systems (NeurIPS 2024).

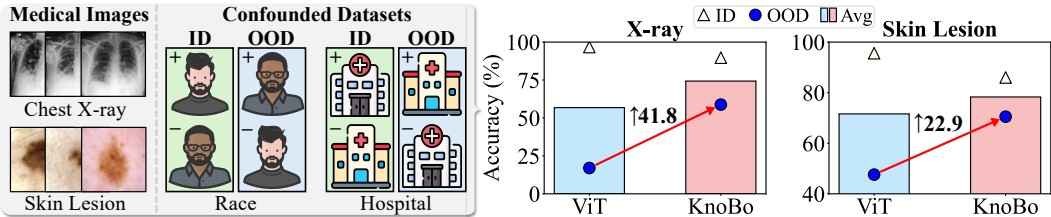

Figure 1: In-domain (ID), out-of-domain (OOD), and average of ID and OOD (Avg) performance on confounded medical image datasets. Our interpretable **Kno**wledge-enhanced **Bo**ttlenecks (**KnoBo**) are more robust to domain shifts (e.g., race, hospital, etc) than fine-tuned vision transformers [17].

Priors are an important signal allowing models to adopt appropriate hypotheses in low or misleading data regimes. We hypothesize that existing visual backbones lack an appropriate prior for robust generalization in medicine. Like previous work identifying that vision backbones have a deep image prior even when entirely untrained [72, 78], we compare the quality of image representations produced by untrained networks on natural versus medical images. Given the output from a frozen untrained visual backbone, we train a linear classifier for predicting a diversity of labels (see Figure 2). Across architecture, these untrained models are higher quality featurizers of natural images than directly using pixels as features. In contrast, **across multiple medical modalities, the deep image prior in current major visual backbones is no more effective than using pixels (and often worse).**

To address the lack of an effective deep image prior for medical images, we propose using an inherently interpretable model design. We draw inspiration from medical education, where students first learn from textbooks and later in a more practical setting during the residency with an attending doctor. Our models mimic this pattern: first, documents are used to identify important knowledge, and then they learn by example from data. We employ concept bottleneck models (CBMs) [41] and enrich them with information derived from resources broadly accessible to medical students. CBMs are a class of inherently interpretable models that factor model decisions into human-readable concepts that are combined linearly. Our methods build on recent approaches for language model (LM) guided bottleneck construction where LMs are prompted for discriminative attributes [90].

We introduce **Kno**wledge-enhanced **Bo**ttlenecks (**KnoBo**) to incorporate knowledge priors that encourage reasoning with factors found in medical documents. KnoBo extends CBMs to medical imaging and employs retrieval-augmented generation into concept design. For example, we extract concepts from medical textbooks as natural language questions like *Is there ground-glass opacity?* to help the model classify whether an X-ray is positive for a respiratory infection. As illustrated in Figure 3, KnoBo factors learning into three parts: (1) an interpretable bottleneck predictor, (2) a prior over the structure of the bottleneck, and (3) a prior over predictor parameters. This factorization allows us to guide the model with a prior rooted in medical documents. The approach relies on an iterative retrieval process where an LM summarizes documents to propose concepts, forming our medical image prior (Sec 4.2). Given the concepts, a pretraining corpus of reports and images is used to construct a classifier for a concept (Sec 4.3). Finally, a CBM is learned using predictions from classifiers on data while regularized by a prior formed from LM generations. (Sec 4.4).

We evaluate KnoBo on our benchmark of confounded tasks. Averaged over confounds, KnoBo increases OOD performance by 41.8% and 22.9% on X-ray and skin lesion datasets, respectively. KnoBo's success in OOD performance comes at little sacrifice in ID settings, providing a better model overall when averaging the two data settings. We also explore 5 different sources of knowledge and reveal that PubMed outperforms other resources in terms of both diversity of information and final prediction performance. Overall, our work demonstrates that a key missing ingredient for robustness to distribution shift in medical imaging models is a prior rooted in knowledge.

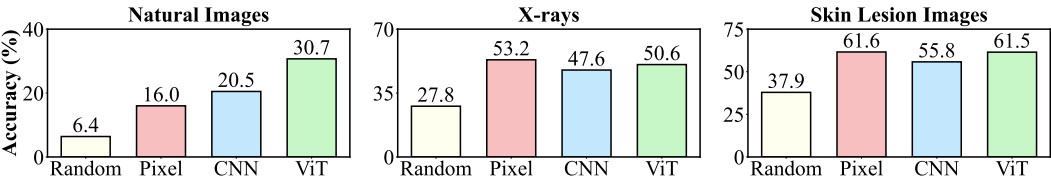

Figure 2: Classification performance on natural and medical images through linear probing using features extracted from untrained and frozen models versus pixels features (See Sec 3 for details).

## 2   Related Work

The rapid advancement in medical foundation models offers the opportunity to develop healthcare AI [55, 49]. However, their lack of transparency presents risks in real-world applications [5]. **Interpretability** is crucial for using models in high-stakes domains [30, 79, 87]. Previous work has primarily focused on post-hoc interpretability [76, 95, 29, 75], which may not provide faithful explanations [69]. As an alternative, inherently interpretable methods produce explanations that align with the model's reasoning processes [8, 3]. In this work, we build upon **Concept Bottleneck Models** (CBMs) [41], which predict by linearly combining human-designed concepts. Recent work [91, 60, 90] scales the applications of CBMs by aligning concepts and images with CLIP [64] and prompting language models to generate concept bottlenecks automatically. In our work, we treat CBMs as the architecture to incorporate knowledge priors for mitigating medical domain shift problems. Our bottlenecks, built from the medical corpus, are attributable and more trustworthy.

**Domain Generalization and Robustness** are critical in medical domains where the distribution of imaging protocols, devices, and patient populations can significantly vary [25]. A line of work studies various domain-shift problems [42, 2], proposing algorithms to learn invariant representations [57, 22, 80, 63] and employing domain/group information for reweighting [71, 93, 44, 96]. However, many studies show those methods do not improve over standard Empirical Risk Minimization (ERM) [67, 26, 33, 27]. Fine-tuning the last layer [68, 40] or selectively fine-tuning a few layers [47] is sufficient for robustness against spurious correlations in those datasets. We address domain shifts in medical imaging from a novel perspective by employing interpretable models to integrate knowledge priors. Our approach encourages models to adhere to diagnostic rules similar to those doctors use rather than relying on spurious correlations. Concurrent work shows bottleneck models can perform well on out-of-domain X-ray data but severely reduced in-domain performance as a consequence [89]. In contrast, we demonstrate a significantly better compromise between OOD and ID performance, using a broader set of modalities and constructing our bottlenecks from medical documents.

**Knowledge Rich Multimodal Reasoning.** Knowledge plays an important role in clinical diagnosis [6]. Some multimodal tasks [82, 53, 74] require models to use explicit outside knowledge to make correct predictions. Previous methods [52, 50, 31] retrieve documents for each example from the external knowledge base as context for models to generate the answer. Our work focuses on leveraging knowledge in medical image classification. **Retrieval-Augmented Generation** [48, 23] has been shown to be beneficial for knowledge-intensive tasks [38], including biomedicine [20, 84]. The retrieved medical documents are either used as context during inference [88] or data for pretraining [94]. In contrast, we treat documents as background knowledge for large language models to build concept bottlenecks. Instead of retrieving documents for every input, we build a global knowledge prior from a medical document corpus, which is shared across all examples.

## 3   Deep Image Priors for Medical Images

This section revisits the concept of deep image priors [72, 78], i.e., some data-agnostic assumptions from model structure, in the context of image classification across various domains. By comparing linear probing using features extracted by untrained deep networks against pixel-based features, we observe that existing vision backbones lack suitable priors for medical domains. This observation motivates our knowledge-enhanced bottlenecks (Sec 4) to integrate more robust priors into models.

**Setup.** Consider a dataset of image-label pairs, $\mathcal{D} = \{(I, y)\}$, where $I$ is an image and $y \in \mathcal{Y}$ denotes the label from one of $N$ classes. The model learns to predict $P(y|I, \theta)$, where $\theta$ is the model parameters. We employ a frozen, untrained vision backbone $\mathcal{V}$ to extract features from $I$, producing a feature vector $\boldsymbol{x} = \mathcal{V}(I)$, where $\boldsymbol{x} \in \mathbb{R}^d$. A linear mapping function $f_\theta : \mathbb{R}^d \to \mathcal{Y}$ is then trained to classify these features into label spaces. In this case, the model parameters $\theta$ will inherit the implicit architectural priors of $\mathcal{V}$. As a baseline, we extract a subset of $d$ pixels directly from the image as the feature without any model-based priors, represented as $\boldsymbol{x}_p \in \mathbb{R}^d$. We compare the classification performance using $\boldsymbol{x}$ versus $\boldsymbol{x}_p$ to probe the efficacy of the vision backbone's priors.

**Experiments.** We evaluate two state-of-the-art vision backbones, ViT-L/14 [17] and ConvNext-L [51], on three categories of images: natural photos (e.g., ImageNet [70]), X-rays (e.g., NIH-CXR [83]), and skin lesion images (e.g., HAM10000 [77]). Each image category has 5 datasets, and we report their average performance in Figure 2 (see Table 11 in the Appendix C.1 for full results).

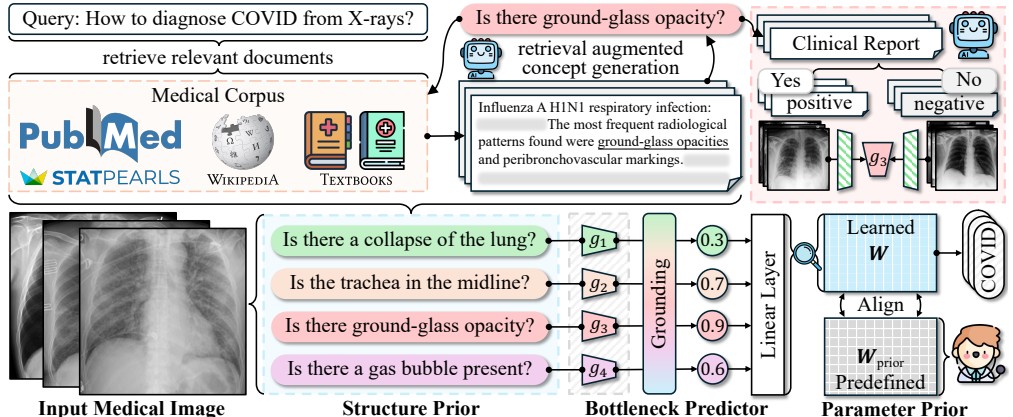

Figure 3: Overview of **Kno**wledge-enhanced **Bo**ttlenecks (**KnoBo**) for medical image classification, comprising three main components: (1) **Structure Prior** (Sec 4.2) constructs the trustworthy knowledge bottleneck by leveraging medical documents; (2) **Bottleneck Predictor** (Sec 4.3) grounds the images onto concepts which are used as input for the linear layer and ; (3) **Parameter Prior** (Sec 4.4) constrains the learning of linear layer with parameters predefined by doctors or LLMs.

Figure 2 (left) shows vision backbones have effective priors for natural images, with ViT notably outperforming pixel by 14.7%. However, pixel features surpass those extracted by vision backbones for specialized domains such as X-ray and skin lesion images. This underscores these deep networks' lack of image priors appropriate for these domains, which can hamper model learning and hurt generalizability. Without guidance from appropriate priors, models can overly rely on data, risking catastrophic failures. We aim to overcome this by injecting additional priors into models.

## 4 Knowledge-enhanced Bottlenecks

In this section, we present **Kno**wledge-enhanced **Bo**ttlenecks (**KnoBo**), a class of CBMs that incorporate knowledge priors that address the failures we identified in Section 3. Figure 3 presents an overview of our method. From left to right, we optimize three terms: (1) **Structure Prior** (Sec 4.2) induces bottleneck structures from medical corpus to incorporate human knowledge as concepts, (2) **Bottleneck Predictor** (Sec 4.3) projects input image onto bottleneck concepts and then feed concept predictions into the linear layer for label prediction, and (3) **Parameter Prior** (Sec 4.4) aligns the learned parameters with known associative information to further enhance priors.

### 4.1 Problem Formulation

**Preliminary on Concept Bottleneck Model.** Given a bottleneck $C$ with $N_C$ concepts, CBMs optimize two functions for predictions: $\hat{y} = f\left(\mathcal{G}(\boldsymbol{x})\right)$, where $\mathcal{G} : \mathbb{R}^d \rightarrow \mathbb{R}^{N_C}$ maps image features into concept space, and $f : \mathbb{R}^{N_C} \rightarrow \mathcal{Y}$ uses concept predictions for final label predictions.

**Formulation**. Our goal is to incorporate priors over $C$, the concept structure, into the learning of the joint probability $\sum_{(I,y) \in \mathcal{D}} \log P\left(y, C, \theta | I\right)$, which can be decomposed into three factors:

$$\log P\left(y, C, \theta | I\right) = \underbrace{\log P\left(C\right)}_{\text{structure prior}} + \underbrace{\log P\left(\theta\right)}_{\text{parameter prior}} + \underbrace{\log P\left(y | I, C, \theta\right)}_{\text{bottleneck predictor}} \tag{1}$$

where we assume the priors over structure $C$ and parameters $\theta$ are independent. The **structure prior** $P(C)$ (Sec 4.2) is formulated as the construction of a bottleneck with $N_C$ concepts, $C = \left\{c_1, c_2, ..., c_{N_C}\right\}$, derived from a background corpus $\mathcal{B}$. Each concept is a factor that humans will use when solving the same task. The **bottleneck predictor** $P\left(y | I, C, \theta\right)$ (Sec 4.3) is a concept bottleneck model that predicts the label conditioned on the input image, bottleneck, and learned parameters. The bottleneck predictor is inherently interpretable, and each parameter in $\theta$ has semantics, denoting the association between concepts and labels. The **parameter prior** $P(\theta)$ (Sec 4.4) regularizes the learning of model parameters $\theta$ with information derived from human knowledge. Jointly optimizing over structure and model parameters is intractable, so we first select a high-quality concept space and then optimize the parameters of the bottleneck jointly with the parameter prior.

## 4.2 Structural Prior

Given a background corpus $\mathcal{B}$ that spans various documents, we aim to identify a bottleneck structure containing concepts beneficial for classifying labels $y \in \mathcal{Y}$. As outlined in Algorithm 1, we use the class names $\mathcal{Y}$ as initial queries to retrieve relevant documents $\mathcal{B}' \subset \mathcal{B}$. Large Language Models (LLMs) are then prompted to generate concepts using these retrieved documents as context: $C' = \text{LLM}(\mathcal{B}')$. These newly generated concepts are added to the bottleneck and used as new queries to retrieve additional documents. This iterative process continues to expand the bottleneck until a predetermined number of concepts $N_C$ is reached. Such concept structures have high likelihood under the language model, conditioned on the background corpus, and the language model probability serves as an implicit prior.

---

**Algorithm 1** Retrieve Augmented Iterative Concept Bottleneck Generation

---

$\mathcal{Y}$, set of target class names
$\mathcal{B}$, set of background documents
$\mathcal{Q} \leftarrow \mathcal{Y}$, class names as initial queries
$C \leftarrow [\,]$ , concepts in the bottleneck
**while** $|C| < N_C$ **do**
  $\mathcal{Q}' \leftarrow [\,]$ , set of new queries
  **for** $q$ in $\mathcal{Q}$ **do**
    $\mathcal{B}' = \textbf{Retrieve}\,(\mathcal{B}, q)$
    $C' = \textbf{LLM}\,(\mathcal{B}')$
    $C \leftarrow C + C', \mathcal{Q}' \leftarrow \mathcal{Q}' + C'$
  **end for**
  $\mathcal{Q} \leftarrow \mathcal{Q}'$ /* *update queries* */
**end while**

---

## 4.3 Bottleneck Predictor

With the structure prior from the background corpus, we optimize (1) the grounding function $\mathcal{G} : \mathbb{R}^d \to \mathbb{R}^{N_C}$, which maps the input image to the concept space, and (2) a linear layer $f : \mathbb{R}^{N_C} \to \mathcal{Y}$ that projects concept predictions onto labels. In practice, we implement $\mathcal{G}$ as a set of grounding functions: $\mathcal{G} = \{g_c\}_{c \in C}$ where each $g_c$ predicts the probability of an image $I$ having the concept $c$, $P(c|I) = g_c(\boldsymbol{x})$, and $\boldsymbol{x} \in \mathbb{R}^d$ is the image feature. Specifically, we derive training examples for grounding functions from a pretraining dataset of image-text pairs. We use the language model to estimate the presence of a concept in the image based on the information in the accompanying text.

**Concept Grounding.** Suppose we have a pretraining dataset $\mathcal{D}_{\text{pre}}$ of image-text pairs $\{(I, t)\}$, where $t$ is a textual description (such as a clinical report) of the image. Based on $t$, we can infer if a concept $c$ is present in the image. This can be automated by prompting a large language model to generate a response indicating whether the text implies the concept. This way, we label our pretraining data as positive and negative examples for each concept $c$, which can be used to train its grounding function. With those annotated training examples, we implement each grounding function as a binary logistic regression classifier: $g_c(\boldsymbol{x}) = \sigma\left(\boldsymbol{x} \cdot \boldsymbol{W}_c^\top\right)$, where $\boldsymbol{W}_c \in \mathbb{R}^d$ is the weights of grounding function and $\sigma$ is the sigmoid activation. Finally, we form a collection of grounding functions $\mathcal{G} = \{g_c\}_{c \in C}$ to map an image feature $\boldsymbol{x}$ into $N_C$ probabilities over all bottleneck concepts, with $\mathcal{G}(\boldsymbol{x}) \in \mathbb{R}^{N_C}$.

**Linear Layer.** Using concept probabilities $\mathcal{G}(\boldsymbol{x})$ as input, we train a simple linear function $f$ to make the final label prediction: $\hat{y} = f\left(\mathcal{G}(\boldsymbol{x})\right) = \mathcal{G}(\boldsymbol{x}) \cdot \boldsymbol{W}^\top$, where $\boldsymbol{W} \in \mathbb{R}^{N \times N_C}$ is the linear weight matrix, with $N$ the number of classes and $N_C$ the number of concepts.

## 4.4 Parameter Prior

The bottleneck predictor is inherently interpretable because the parameters of the linear layer encode the affinity between labels and concepts. Therefore, we can guide the parameters based on prior knowledge, i.e., if the label $y$ is positively related to concept $c$ based on background knowledge, the weight $w_{y,c} \in \boldsymbol{W}$ should be high. We hope the learned parameters do not deviate too much from this assumption, otherwise, the model may capture spurious correlations in the data.

To enforce this, we let language models define a weight matrix of priors $\boldsymbol{W}_{\text{prior}} \in \mathbb{R}^{N \times N_C}$, with each element $w_{y,c} \in \{-1, +1\}$ indicating the sign of a preferred correlation between the label $y$ and concept $c$. The prior loss is calculated as the L1 distance between between $\boldsymbol{W}$ and $\boldsymbol{W}_{\text{prior}}$:

$$\mathcal{L}_{\text{prior}} = \frac{1}{N \cdot N_C} \cdot ||\tanh\left(\boldsymbol{W}\right) - \boldsymbol{W}_{\text{prior}}||_1 \tag{2}$$

in which we apply tanh activation on $\boldsymbol{W}$ to scale the linear weights to $(-1, 1)$, matching the scale of the weights in the prior matrix. This adjustment aligns the model's parameters with the expected sign of the correlations based on prior knowledge. The final loss function to train the linear layer is the sum of the cross-entropy loss and the prior loss: $\mathcal{L} = \mathcal{L}_{\text{CE}} + \mathcal{L}_{\text{prior}}$.

In summary, we search for a structure $C$ that is consistent with prior knowledge from a background corpus $\mathcal{B}$ to severe as the bottleneck for the predictor $P(y|I, C, \theta)$. The parameters $\theta$ are aligned with the predefined correlations between labels and concepts identified by language models.

## 5 Experimental Setup

This section introduces (1) the confounded and unconfounded medical datasets to evaluate the robustness of our knowledge-enhanced bottlenecks (Sec 5.1), (2) the black-box and interpretable baselines for comparison (Sec 5.2), and (3) the implementation details of our method (Sec 5.3).

### 5.1 Datasets

We evaluate two groups of datasets for each modality: (1) the **confounded datasets**, which aim to assess the **robustness** of models by creating splits with spurious correlations; (2) the **unconfounded datasets** are randomly split to measure the models' **performance** in natural settings.

**Confounded Datasets.** As illustrated on the left of Figure 1, we formulate the confounded datasets as binary classification tasks, where each class is confounded with one factor. The confounding combinations are reversed for in-domain (train and validation), and out-of-domain (test) splits.

The confounded datasets of **chest X-ray** are constructed from NIH-CXR [83] and CheXpert [35] with their provided attributes: (1) **NIH-sex** uses sex (male, female) as the confounding factor; (2) **NIH-age** confounds the data with age (young, old); (3) **NIH-pos** analyzes the patient's position (standing, lying down) during X-ray examinations; (4) **CheXpert-race** splits the data based on patient's race (white, black or African American); (5) **NIH-CheXpert** confounds X-rays across datasets (NIH, CheXpert).

The confounded datasets of **skin lesion** are derived from the International Skin Imaging Collaboration (ISIC): (1) **ISIC-sex** and (2) **ISIC-age** are set up similarly to the X-ray datasets mentioned previously; (3) **ISIC-site** studies lesions developed on different sites of the body (head, extremities); (4) **ISIC-color** evaluates examples with different skin colors (light, dark); and (5) **ISIC-hospital** uses instances sampled from hospitals in different cities (Barcelona, Vienna).

**Unconfounded datasets**. We evaluate 10 datasets with random splits, 5 for each modality. *X-ray*: **Pneumonia** [39], **COVID-QU** [9], **NIH-CXR** [83], **Open-i** [16], and **VinDr-CXR** [58]. *Skin Lesion*: **HAM10000** [77], **BCN20000** [14], **PAD-UFES-20** [62], **Melanoma** [36], and **UWaterloo** [45].

All datasets are split into train/validation/test and ensure the validation and test set are balanced across classes. Detailed statistics and additional information on each dataset are provided in Appendix A.

**Pretraining Datasets.** The training of vision backbones and concept grounding functions utilizes datasets with image-text pairs. For X-rays, we choose MIMIC-CXR [37], which contains 377,110 X-ray images with accompanying clinical reports. Since there is no existing text-annotated dataset for skin lesion images, we employ GPT-4V [61] to generate captions (see examples in Figure 9) for a subset of 56,590 images from ISIC, without overlap of the confounded and unconfounded datasets.

### 5.2 Baselines

We compare KnoBo against both black-box models and interpretable concept bottleneck models.

**Black-box Models.** We include two end-to-end fine-tuning baselines: (1) **ViT-L/14** [17] and (2) **DenseNet121** [32], both pretrained on the pretraining datasets mentioned earlier. Additionally, (3) **Linear Probe** extracts visual features with the frozen ViT-L/14 encoder and learns a linear layer for classification. (4) **Language-shaped Learning** (LSL) [56] aims to disentangle the impact of knowledge and interpretable structure. Inspired by LSL via captioning, we finetune a ViT-L/14 with the same data used for concept grounding functions and apply a linear layer (see Appendix B.2).

**Concept Bottleneck Models.** (1) **Post-hoc CBM** (PCBM-h) [91] ensembles concept bottleneck models with black-box residual predictors. We let PCBM-h use the same bottlenecks as our KnoBo method; (2) **LaBo** [90] applies language models to generate concepts, followed by the submodular selection to identify a subset that enhances performance. Following their original settings, PCBM-h and LaBo use CLIP (fine-tuned on medical pretraining datasets) to align concepts with images.

| Method | NIH-sex | | | NIH-age | | | NIH-pos | | | CheXpert-race | | | NIH-CheXpert | | |
|---|---|---|---|---|---|---|---|---|---|---|---|---|---|---|---|
| | ID | OOD | Avg | ID | OOD | Avg | ID | OOD | Avg | ID | OOD | Avg | ID | OOD | Avg |
| ViT-L/14 | **97.0** | 30.9 | 64.0 | **97.4** | 3.2 | 50.3 | **99.7** | 2.7 | 51.2 | **89.4** | 48.2 | 68.8 | **99.9** | 0.1 | 50.0 |
| DenseNet | 91.4 | 32.1 | 61.8 | 90.6 | 15.6 | 53.1 | 99.3 | 1.0 | 50.2 | 85.0 | 55.4 | 70.2 | **99.9** | 0.2 | 50.1 |
| Linear Probe | 94.2 | 46.7 | 70.5 | 95.0 | 11.4 | 53.2 | 99.3 | 17.0 | 58.2 | 87.8 | 71.4 | 79.6 | 99.6 | 6.8 | 53.2 |
| LSL | 84.0 | 74.3 | 79.2 | 79.8 | **53.8** | 66.8 | 95.3 | 39.0 | 67.2 | 80.4 | 76.4 | 78.4 | 95.0 | 31.8 | 63.4 |
| PCBM-h | 94.2 | 45.6 | 69.9 | 95.0 | 10.8 | 52.9 | 99.3 | 17.0 | 58.2 | 88.0 | 71.4 | 79.7 | 99.6 | 8.2 | 53.9 |
| LaBo | 91.4 | 51.3 | 71.4 | 92.8 | 14.4 | 53.6 | 98.0 | 24.3 | 61.2 | 86.8 | 69.2 | 78.0 | 98.4 | 14.9 | 56.7 |
| KnoBo (ours) | 88.6 | 78.6 | 83.6 | 88.8 | 38.8 | 63.8 | 95.7 | **45.3** | 70.5 | 84.0 | 79.0 | 81.5 | 91.6 | **52.3** | 72.0 |

| Method | ISIC-sex | | | ISIC-age | | | ISIC-site | | | ISIC-color | | | ISIC-hospital | | |
|---|---|---|---|---|---|---|---|---|---|---|---|---|---|---|---|
| | ID | OOD | Avg | ID | OOD | Avg | ID | OOD | Avg | ID | OOD | Avg | ID | OOD | Avg |
| ViT-L/14 | **92.0** | 69.0 | 80.5 | **95.0** | 61.3 | **78.2** | **94.8** | 38.3 | 66.6 | **96.9** | 59.2 | 78.1 | 99.2 | 10.0 | 54.6 |
| DenseNet | 85.3 | 76.0 | 80.7 | 93.7 | 61.3 | 77.5 | 81.7 | 54.5 | 68.1 | 93.9 | 44.6 | 69.2 | 98.4 | 15.1 | 56.8 |
| Linear Probe | 86.0 | 69.7 | 77.8 | 92.7 | 60.7 | 76.7 | 90.2 | 37.2 | 63.7 | 90.8 | 65.8 | 78.3 | **100.0** | 27.1 | 63.6 |
| LSL | 82.7 | 78.3 | 80.5 | 90.3 | 66.0 | 78.2 | 84.3 | 50.2 | 67.3 | 87.3 | 73.1 | 80.2 | 99.6 | 27.9 | 63.8 |
| PCBM-h | 86.7 | 69.0 | 77.8 | 93.0 | 59.3 | 76.2 | 90.0 | 38.5 | 64.3 | 91.2 | 66.5 | 78.9 | **100.0** | 26.8 | 63.4 |
| LaBo | 83.0 | 69.3 | 76.2 | 91.3 | 61.0 | 76.2 | 88.0 | 39.3 | 63.7 | 86.9 | 78.9 | 82.9 | **100.0** | 8.6 | 54.3 |
| KnoBo (ours) | 84.0 | 79.7 | 81.8 | 88.0 | 67.7 | 77.8 | 80.7 | **58.8** | 69.8 | 89.2 | 75.8 | 82.5 | 88.2 | **77.5** | 82.9 |

Table 1: Results on 10 **confounded datasets** of two modalities (top-5 are X-ray and bottom-5 are skin lesion). We report in-domain (ID), out-of-domain (OOD), and average of ID and OOD (Avg) accuracy. The best score of each column is **bold**, and the second best is underlined.

All baselines use backbones trained on the same pretraining data as our method to ensure a fair comparison. Appendix B.2 provides additional details about the baselines.

**Evaluation Metrics.** We use accuracy as the metric since all evaluated datasets are single-label classification tasks with balanced validation and test sets. For confounded datasets, we report in-domain (ID, validation), out-of-domain (OOD, test), and domain-average (mean of ID and OOD) accuracies, along with domain gaps ($\Delta = |\text{ID} - \text{OOD}|$), where a lower $\Delta$ indicates better robustness. For unconfounded datasets, we report test accuracy. A robust and performant model must achieve a good compromise between confounded and unconfounded datasets. For all the baselines and our KnoBo method, the checkpoints with the highest validation accuracy are evaluated on the test set.

### 5.3 Implementation Details

**Pretraining of Medical CLIP.** We fine-tune OpenCLIP [34] (ViT-L/14 pretrained on LAION-2B [73]) on the pretraining medical data for each modality. Unlike previous work [18, 85, 92] that directly pairs medical images with sentences from clinical reports, we preprocess the reports by employing GPT-4 [1] to extract short phrases. Our CLIP models perform the best for both X-ray and skin lesion datasets in zero-shot and linear probing, as shown in Table 9 in the Appendix.

**Medical Corpus.** We download 5.5 million articles from PubMed and segment them into 156.9 million snippets to serve as documents for retrieval. Alternatively, we take the medical corpus organized by MEDRAG [88], including documents from Wikipedia, StatPearls, and medical textbooks. We employ BM25 [66] as the ranking function for document retrieval.

**KnoBo Details.** We select GPT-4 (`gpt-4-0613`) as the underlying LLM for retrieval-augmented concept generation (Sec 4.2). For training concept grounding functions (Sec 4.3), we opt for Flan-T5-XXL [10] to annotate clinical reports for each concept, considering cost-efficiency. Unless otherwise specified, KnoBo uses bottlenecks constructed from PubMed, each with 150 concepts. Figure 5 shows our prompt, and during concept generation, we apply several heuristic filters (Appendix B.3).

## 6 Results

In this section, we discuss KnoBo's performance on confounded and unconfounded medical image datasets (Sec 6.1) and analyze different knowledge resources and our model design (Sec 6.2).

| Method | Chest X-ray Datasets | | | | | | Skin Lesion Datasets | | | | | |
|---|---|---|---|---|---|---|---|---|---|---|---|---|
| | ID | OOD | $\Delta \downarrow$ | Avg | Unconfd | Overall | ID | OOD | $\Delta \downarrow$ | Avg | Unconfd | Overall |
| ViT-L/14 | **96.7** | 17.0 | 79.7 | 56.8 | 70.2 | 63.5 | **95.6** | 47.6 | 48.0 | 71.6 | **84.3** | 77.9 |
| DenseNet | 93.2 | 20.9 | 72.4 | 57.1 | 66.0 | 61.5 | 90.6 | 50.3 | 40.3 | 70.4 | 71.0 | 70.7 |
| Linear Probe | 95.2 | 30.7 | 64.5 | 62.9 | 73.8 | 68.4 | 91.9 | 52.1 | 39.8 | 72.0 | 82.8 | 77.4 |
| LSL | 86.9 | 55.1 | 31.8 | 71.0 | 67.0 | 69.0 | 88.9 | 59.1 | 29.8 | 74.0 | 77.2 | 75.6 |
| PCBM-h | 95.2 | 30.6 | 64.6 | 62.9 | **74.7** | 68.8 | 92.2 | 52.0 | 40.1 | 72.1 | 81.7 | 76.9 |
| LaBo | 93.5 | 34.8 | 58.7 | 64.2 | 72.1 | 68.1 | 89.9 | 51.4 | 38.4 | 70.6 | 80.0 | 75.3 |
| KnoBo (ours) | 89.7 | **58.8** | **30.9** | **74.3** | 73.1 | **73.7** | 86.0 | **70.5** | **14.1** | **78.3** | 78.1 | **78.2** |

Table 2: Averaged results across all datasets, including in-domain (ID), out-of-domain (OOD), domain-gap ($\Delta$, lower is better), and mean of ID and OOD (Avg) accuracy for confounded datasets. For unconfounded datasets (Unconfd), we report test accuracy. Overall performance is calculated as the mean of the Avg and Unconfd, the overall tradeoff between data conditions.

| Knowledge Source | Chest X-ray Datasets | | | | Skin Lesion Datasets | | | |
|---|---|---|---|---|---|---|---|---|
| | Confd | Unconfd | Overall | Diversity | Confd | Unconfd | Overall | Diversity |
| PROMPT | 72.9 | 72.8 | 72.9 | 0.542 | **78.4** | 77.0 | 77.7 | 0.332 |
| TEXTBOOKS | 72.0 | 72.9 | 72.4 | 0.585 | 77.5 | 78.3 | 77.9 | 0.350 |
| WIKIPEDIA | 72.8 | 72.7 | 72.8 | 0.542 | 77.6 | 77.9 | 77.8 | 0.356 |
| STATPEARLS | 73.4 | 72.0 | 72.7 | 0.598 | 77.1 | **79.1** | 78.1 | **0.379** |
| PUBMED | **74.3** | **73.1** | **73.7** | **0.619** | 78.3 | 78.1 | **78.2** | 0.341 |

Table 3: Comparison of concept bottlenecks built from different knowledge sources. PROMPT is our baseline without retrieving documents for concept generation. We report the accuracy of confounded (Confd, average over ID and OOD), unconfounded (Unconfd) datasets, and the overall performance of all datasets. Diversity measures the difference between the concepts in a bottleneck.

## 6.1 Main Results

**KnoBo is more robust to domain shifts.** Table 1 shows the results on 10 confounded datasets of X-ray and skin lesions. Black-box models excel at in-domain (ID) data but drop significantly on out-of-domain (OOD) data, especially in datasets confounded by hospitals/resources (NIH-CheXpert and ISIC-hospital), which can be common when collecting medical datasets [12, 81]. KnoBo outperforms baselines in OOD and domain-average accuracy by large margins, ranking top-1 in eight datasets and second-best in the other two. End-to-end models (ViT-L/14, DenseNet) exhibit larger domain gaps than linear probes, as they have more parameters to optimize performance on in-domain data and capture spurious correlations. Shaping the visual representations with knowledge (LSL) improves robustness but underperforms KnoBo, with lower ID, OOD, and average performance across most datasets. PCBM-h combines interpretable and black-box predictions but exhibits behaviors similar to black-box models with severe drops across domains. Unlike KnoBo, which uses medical documents to create one global bottleneck for each modality, LaBo builds a bottleneck for each dataset using the in-domain data, which can be biased and affected by confounding factors and performs more poorly. In summary, KnoBo mitigates the catastrophic failures in domain shifts encountered by black-box and is more robust against various confounding factors across modalities.

**KnoBo performs the best across confounded and unconfounded data.** Table 2 illustrates the performance averaged across confounded and unconfounded datasets. For both types of medical images, KnoBo achieves the best out-of-domain (OOD) and domain-average performance (Avg) with minimal domain gaps ($\Delta$), outperforming the strongest end-to-end baseline (ViT-L/14) by 41.8% (X-ray) and 22.9% (skin lesion) in OOD accuracy. KnoBo achieves competitive performance for unconfounded X-ray datasets, trailing the best-performing black-box model (Linear Probe) by only 0.7%. While KnoBo is less competitive on skin lesion datasets due to the lack of large-scale pretraining data for accurate concept grounding, it still maintains performance comparable to the baselines. By calculating the mean accuracy across both confounded and unconfounded datasets, KnoBo ranks top across all models, confirming that our knowledge-enhanced, interpretable approach is a promising direction for building more robust and performant systems for medical imaging.

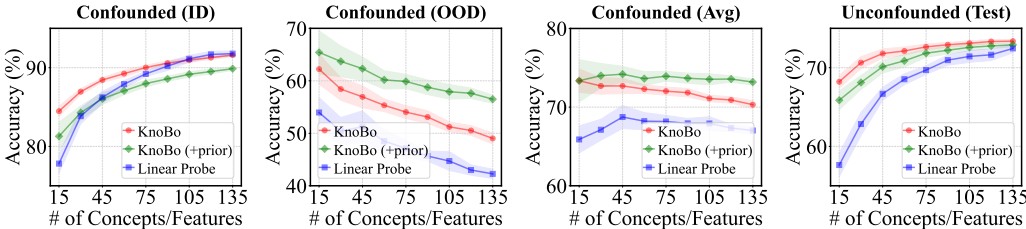

Figure 4: Ablation of **bottleneck sizes** on X-ray datasets. The x-axis is the number of randomly selected concepts (KnoBo) or visual features (Linear Probe). +prior means adding parameter prior.

| Method | Chest X-ray Datasets | | | | | | Skin Lesion Datasets | | | | | |
|---|---|---|---|---|---|---|---|---|---|---|---|---|
| | ID | OOD | $\Delta\downarrow$ | Avg | Unconfd | Overall | ID | OOD | $\Delta\downarrow$ | Avg | Unconfd | Overall |
| KnoBo | 89.7 | **58.8** | **30.9** | **74.3** | 73.1 | **73.7** | 86.0 | **70.5** | 14.1 | **78.3** | 78.1 | **78.2** |
| w/o $\mathcal{G}$ | 87.8 | 51.5 | 36.3 | 69.6 | 70.1 | 69.9 | 83.7 | 69.4 | **11.5** | 76.6 | 70.2 | 73.4 |
| w/o $\mathcal{L}_{\text{prior}}$ | 91.6 | 48.1 | 43.5 | 69.8 | **73.6** | 71.7 | **86.5** | 69.1 | 16.6 | 77.8 | **78.4** | 78.1 |

Table 5: Ablation studies on concept grounding ($\mathcal{G}$; Sec 4.3) and parameter prior ($\mathcal{L}_{\text{prior}}$; Sec 4.4).

## 6.2 Analysis

In this section, we compare the bottlenecks constructed from different knowledge sources. We evaluate the impact of each KnoBo component on the final performance, including bottleneck size, concept grounding function, and parameter prior. Additional analyses are available in Appendix C.

**Knowledge Sources.** Besides the empirical results on confounded and unconfounded datasets, we measure the diversity of bottleneck $C$ as **Diversity**$(C) = \frac{1}{|C|^2 - |C|} \sum_{c_i \in C} \sum_{c_j \in C}^{i \neq j} \left(1 - \text{sim}(c_i, c_j)\right)$, where the $\text{sim}(\cdot)$ is the cosine similarity of concept features encoded by sentence transformer [65]. The **Diversity** computes the distance between each concept and every other concept in the bottleneck. Table 3 compares different knowledge sources. The retrieval-augmented bottlenecks perform better than those generated by prompting, especially for skin lesions, where more specific knowledge is required because prompting lacks diversity. Across both modalities, PubMed is the best overall, performing better for the X-ray modality than other knowledge sources and among the best for skin lesion modalities. Moreover, shown in Table 6, our retrieval-augmented concepts are attributable, which allows doctors to verify the source of knowledge.

**Human Evaluation on Bottlenecks.** In evaluations by two medical students, information from all knowledge sources is rated as highly relevant and groundable. Two medical students evaluated the quality of bottlenecks using two metrics: (1) **Relevance** measures the concept's relevance to diagnosing diseases on a scale from 1 (not at all relevant) to 4 (mostly relevant), and (2) **Groundability** assesses the verifiability of the concept from the image on a scale from 1 to 4. We evaluated 30 randomly sampled concepts from each bottleneck. Table 4 presents these metrics for bottlenecks constructed from five different knowledge sources. While all bottlenecks show good relevance, groundability scores are lower, reflecting the challenge of deriving visual concepts from text-only data.

Table 4: Relevance and Groundability of concepts in bottlenecks generated from different resources, as evaluated by student doctors.

| Knowledge Source | Relevance | | Groundability | |
|---|---|---|---|---|
| | X-ray | Skin | X-ray | Skin |
| PROMPT | 3.83 | 3.93 | 3.03 | 3.00 |
| TEXTBOOKS | 3.70 | 3.80 | 2.90 | 3.27 |
| WIKIPEDIA | 3.80 | 3.67 | 2.83 | 3.33 |
| STATPEARLS | 3.87 | 3.80 | 2.70 | 2.97 |
| PUBMED | 3.70 | 3.83 | 2.77 | 3.20 |

**Bottleneck Size.** Figure 4 compares KnoBo and linear probes while varying the number of concepts/features. KnoBo consistently outperforms linear probes across all metrics when given the same quota of features, and KnoBo can obtain better performance with fewer features. This indicates that interpretable concept scores have more effective priors than black-box visual features.

**Ablations.** Table 5 summarizes experiments ablating major components of our approach. Row 2 shows the performance of using dot-products from prompted CLIP models as concepts, which

| Bottleneck | Concept | Query | Reference Document |
|---|---|---|---|
| X-ray (PubMed) | Is there lung collapse? | Atelectasis | Atelectasis and pneumonia were diagnosed on radiological and clinical criteria. **Atelectasis was diagnosed when a finding of lung collapse was made on chest X-ray**, chest CT and/or lung ultrasound. [Source] |
| X-ray (StatPearls) | Is there a widened mediastinum on chest X-ray? | Aortic Enlargement | On chest x-ray (CXR), **findings that may indicate aortic pathology include a widened mediastinum**, loss of the aortic knob contour, inferiorly displaced left bronchus, and left pleural effusion. [Source] |
| Skin (Textbook) | Does the lesion have a waxy appearance? | Seborrheic Keratosis | Lesions have no malignant potential but may be a cosmetic problem. **Present as exophytic, waxy brown papules and plaques with prominent follicle openings.** [In the book: First Aid Step-2] |
| Skin (Wikipedia) | Are there small blood vessels running over the skin lesion? | Basal Cell Carcinoma | BCC, also known as basal-cell cancer, is the most common type of skin cancer. It often appears as a painless raised area of skin, **which may be shiny with small blood vessels running over it**. [Source] |

Table 6: We show concepts from various bottlenecks by image modality (corpus), with the queries for retrieval and the corresponding reference documents to generate the concept. Every concept is attributable, allowing medical professionals to verify its origin in the supporting documentation.

markedly reduces performance. This shows the importance of knowledge grounding in ensuring KnoBo's effectiveness. However, this step can be simplified as more advanced medical foundation models are available. Row 3 shows performance omitting the parameter prior. It is an important mechanism for constraining the final learning phase, resulting in consistent OOD improvements. This is also reflected in Figure 4, where with the parameter prior, KnoBo performs better on OOD splits while decreasing on in-domain as the parameter prior constrains the model to rely on data.

## 7   Conclusion and Limitation

In this paper, we analyze domain-shift problems in medical image analysis and identify a missing medical deep image prior as a main contributor to poor performance. To address this, we introduce knowledge-enhanced bottlenecks (KnoBo) to integrate knowledge priors from medical documents. Across two medical image modalities under various domain shifts, KnoBo significantly improves robustness over black-box baselines.

**Limitation.** KnoBo assumes the availability of medical multimodal datasets, limiting applications to rare conditions. While our work improves robustness, medical experts do not fail in these ways, and they should be used in conjunction with models. Our preliminary work with PubMed suggests it is a valuable resource for developing medical models, and future research can explore how to utilize such fruitful knowledge resources more effectively.

## Acknowledgement

This study was conducted at the University of Pennsylvania School of Engineering and Applied Science and supported in part by the Office of the Director of National Intelligence (ODNI), Intelligence Advanced Research Projects Activity (IARPA), via the HIATUS Program contract #2022-22072200005, and gifts from the UPenn ASSET center and Ai2. The views and conclusions contained herein are those of the authors and should not be interpreted as necessarily representing the official policies, either expressed or implied, of ODNI, IARPA, or the U.S. Government. The U.S. Government is authorized to reproduce and distribute reprints for governmental purposes, notwithstanding any copyright annotation therein. Michael S. Yao was supported by the National Institutes of Health (F30 MD020264). James C. Gee was also supported by the NIH (R01 EB031722).

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

# A  Dataset

| Dataset Name | Confounding Factor | n. of Class | Class Names | n. of Images | | |
|---|---|---|---|---|---|---|
| | | | | train | val | test |
| NIH-sex | sex | 2 | Atelectasis, Effusion | 3000 | 500 | 1000 |
| NIH-age | age | 2 | No finding, Has findings | 3000 | 500 | 500 |
| NIH-pos | position | 2 | Atelectasis, Effusion | 3000 | 300 | 300 |
| CheXpert-race | race | 2 | No finding, Has findings | 4000 | 500 | 500 |
| NIH-CheXpert | dataset | 2 | Atelectasis, Effusion | 5000 | 1000 | 1000 |
| Pneumonia [39] | - | 2 | Normal, Pneumonia | 5216 | 16 | 624 |
| COVID-QU [9] | - | 4 | COVID, Lung Opacity, Normal, Viral Pneumonia | 16930 | 2115 | 2114 |
| NIH-CXR [83] | - | 6 | Atelectasis, Cardiomegaly, Effusion Consolidation, Edema, Pneumonia | 8066 | 1140 | 2317 |
| Open-i [16] | - | 3 | Cardiomegaly, Lung Opacity, Normal | 884 | 438 | 890 |
| VinDr-CXR [58] | - | 7 | Aortic enlargement, Cardiomegaly, Pulmonary fibrosis, Lung Opacity, Pleural thickening, Nodule, Normal | 1400 | 175 | 175 |

Table 7: Detailed statistics of the 10 Chest X-ray datasets evaluated in this work.

| Dataset Name | Confounding Factor | n. of Class | Class Names | n. of Images | | |
|---|---|---|---|---|---|---|
| | | | | train | val | test |
| ISIC-sex | sex | 2 | Benign, Malignant | 2400 | 300 | 300 |
| ISIC-age | age | 2 | Benign, Malignant | 2800 | 300 | 300 |
| ISIC-site | body site | 2 | Benign, Malignant | 2000 | 600 | 600 |
| ISIC-color | skin color | 2 | Benign, Malignant | 1800 | 260 | 260 |
| ISIC-hospital | hospital | 2 | Benign, Malignant | 2400 | 280 | 280 |
| HAM10000 [77] | - | 7 | Actinic Keratoses, Basal Cell Carcinoma, Benign Keratosis-like Lesions, Dermatofibroma, Melanocytic Nevi, Melanoma, Vascular Lesions | 8010 | 1000 | 1000 |
| BCN20000 [14] | - | 4 | Nevus, Basal Cell Carcinoma Melanoma, Actinic/Seborrheic Keratosis | 4800 | 800 | 800 |
| PAD-UFES-20 [62] | - | 2 | Basal/Squamous Cell Carcinoma, Actinic/Seborrheic Keratosis | 1602 | 200 | 200 |
| Melanoma [36] | - | 2 | Benign, Malignant | 8605 | 1000 | 1000 |
| UWaterloo [45] | - | 2 | melanoma, not melanoma | 166 | 20 | 20 |

Table 8: Detailed statistics of the 10 Skin Lesion datasets evaluated in this work.

Table 7 and 8 show the detailed statistics for all 20 datasets evaluated in this paper, where we list the number of classes with class names, the number of images for training, validation, and testing. For confounded datasets, here we explain some details about how we define the confounding factors:

- **NIH-sex** and **ISIC-sex** are built based on the patient sex from NIH-CXR [83] and ISIC.

- **NIH-age** defines young as patient's age $\leq 30$ and old as patient's age $\geq 60$.

- **NIH-pos** refers to the patient's position during an X-ray procedure. The standard position, Posterior Anterior (PA), involves the patient standing, while the Anterior-Posterior (AP) view is used when the patient cannot stand and must lie down.

- **CheXpert-race** selects racial subgroups (White, Black or African American) from CheXpert [35].

- **NIH-CheXpert** confounds classes by sourcing X-rays from either the NIH-CXR or CheXpert datasets. Specifically, data for one disease is obtained from one dataset, while data for a different disease is sourced from the other dataset.

| Model | Pneumonia | | COVID-QU | | NIH-CXR | | Open-i | | VinDr-CXR | | Average | |
|---|---|---|---|---|---|---|---|---|---|---|---|---|
| | ZS | LP | ZS | LP | ZS | LP | ZS | LP | ZS | LP | ZS | LP |
| Random | 50.0 | 50.0 | 25.0 | 25.0 | 16.7 | 16.7 | 33.0 | 33.0 | 14.3 | 14.3 | 27.8 | 27.8 |
| DenseNet | - | 83.8 | - | 82.9 | - | 53.0 | - | 63.8 | - | 27.4 | - | 62.2 |
| OpenAI-CLIP | 62.5 | 82.4 | 6.7 | 91.0 | 35.0 | 49.6 | 21.6 | 60.3 | 16.0 | 33.1 | 28.4 | 63.3 |
| OpenCLIP | 62.5 | 77.4 | 6.3 | 90.0 | 7.5 | 47.9 | 21.6 | 58.7 | 15.4 | 34.9 | 22.7 | 61.8 |
| PubMedCLIP | 63.3 | 72.9 | 22.0 | 87.7 | 30.4 | 47.7 | 26.4 | 59.6 | 15.4 | 26.9 | 31.5 | 58.9 |
| BioMedCLIP | 74.0 | 85.4 | 11.8 | 90.3 | 31.4 | 58.7 | 56.0 | 67.6 | 22.3 | 36.6 | 39.1 | 67.7 |
| PMC-CLIP | 57.7 | 84.9 | 48.1 | 94.9 | 40.4 | 60.6 | 57.6 | 67.4 | 16.0 | 42.3 | 43.9 | 70.0 |
| MedCLIP | **84.9** | 89.9 | **68.6** | 87.4 | 25.1 | 64.8 | **70.2** | 71.9 | 24.6 | 40.0 | 54.7 | 70.8 |
| Ours | 77.7 | 88.6 | 60.6 | 94.9 | **47.4** | 68.4 | 67.5 | 73.3 | **29.1** | **44.0** | **56.5** | 73.8 |
| − Extraction | 57.2 | 88.8 | 46.0 | **95.3** | 41.4 | 68.2 | 62.9 | 72.1 | 21.1 | 46.0 | 45.7 | **74.3** |

| Model | HAM10000 | | BCN20000 | | PAD-UFS-20 | | Melanoma | | UWaterloo | | Average | |
|---|---|---|---|---|---|---|---|---|---|---|---|---|
| | ZS | LP | ZS | LP | ZS | LP | ZS | LP | ZS | LP | ZS | LP |
| Random | 14.3 | 14.3 | 25.0 | 25.0 | 50.0 | 50.0 | 50.0 | 50.0 | 50.0 | 50.0 | 37.9 | 37.9 |
| DenseNet | - | 79.0 | - | 69.6 | - | 69.5 | - | 91.9 | - | 45.0 | - | 71.0 |
| OpenAI-CLIP | 3.6 | 79.9 | 28.3 | 67.9 | 47.0 | 84.5 | 50.9 | 91.6 | 50.0 | 60.0 | 35.9 | 76.8 |
| OpenCLIP | 5.2 | 82.2 | 25.0 | 67.8 | 45.0 | 83.5 | 50.1 | 92.6 | 55.0 | 80.0 | 36.1 | 81.2 |
| PubMedCLIP | 4.8 | 76.3 | 28.9 | 64.4 | 50.5 | 85.5 | 48.8 | 92.2 | 50.0 | 60.0 | 36.6 | 75.7 |
| BioMedCLIP | 60.4 | 75.2 | 27.5 | 61.8 | **61.0** | 84.5 | 57.3 | 90.0 | 50.0 | 65.0 | 51.2 | 75.3 |
| PMC-CLIP | 25.1 | 82.4 | 24.8 | 67.6 | 55.0 | 86.0 | 66.1 | 92.7 | 55.0 | 55.0 | 45.2 | 76.7 |
| MedCLIP | 8.5 | 71.4 | 22.6 | 55.1 | 50.0 | 71.0 | 50.1 | 89.9 | 50.0 | 50.0 | 36.2 | 67.5 |
| Ours | **61.5** | 82.9 | **53.0** | 71.0 | 56.5 | 86.5 | **84.0** | 93.5 | **75.0** | **80.0** | **66.0** | **82.8** |
| − Extraction | 50.9 | **83.3** | 46.5 | **72.0** | 52.0 | **86.5** | 80.1 | **96.0** | 70.0 | 70.0 | 59.9 | 81.6 |

Table 9: Zero-shot (ZS) and Linear Probe (LP) results of different models on five chest X-ray and five skin lesion datasets (not confounded, random split). The best score is **bold**, and the second best is underlined. − Extraction stands for not using LLM to extract findings from clinical reports.

- **ISIC-age** thresholds young as patient's age $\leq 30$ and old as patient's age $\geq 70$.
- **ISIC-site** focuses on lesions located either on the head or an extremity.
- **ISIC-color** organizes images based on the Fitzpatrick scale of skin tones [19]. Fitzpatrick I is classified as light skin, and III, IV, and V as dark skin, according to the annotations from [4].
- **ISIC-hospital** introduces confounds in classes by using lesion images exclusively from either the Hospital Clínic de Barcelona or the Medical University of Vienna.

# B  Implementation Details

## B.1  CLIP Pretraining

We experimented with existing CLIP models in the medical domain and found their performance to be unreliable. Therefore, we decide to train our own CLIP models for X-ray and skin lesion images.

**Pretraining Dataset.** For X-rays, we utilize the MIMIC-CXR dataset [37], specifically selecting only the PA and AP X-rays, which results in 243,334 images, each accompanied by a clinical report written by doctors. For Skin Lesion images, we employ the ISIC dataset and use GPT-4V [61] to generate clinical reports for 56,590 images, examples are shown in Figure 9. We preprocess these reports by extracting medically relevant findings, each described in a short and concise term. The example below demonstrates a report alongside the findings captured by GPT-4. In total, we assemble 953K image-text pairs for X-rays and 438K for skin lesion images.

> **FINAL REPORT** HISTORY: Unresponsive. Evaluate for pneumonia.
> COMPARISON: Chest radiographs ___ and ___. CT thoracic spine ___.
> FINDINGS: Portable frontal view of the chest. The lung volumes are low. No pleural effusion or pneumothorax. There is bibasilar atelectasis, left greater than right. Heart size is normal. Mediastinal and hilar structures are unremarkable. The configuration of the trachea is unchanged from prior cross-sectional imaging.
> IMPRESSION: Low lung volumes without an acute cardiopulmonary process.
> **FINDINGS (GPT-4)**: low lung volumes, bibasilar atelectasis, left greater than right, normal heart size, trachea unchanged

**Training Details.** We utilize the training script from OpenCLIP [34] and select ViT-L/14 as the backbone. Training is performed on 4 RTX A6000 GPUs for 10 epochs with a batch size of 128 and a learning rate of $1e^{-5}$. We choose checkpoints based on the lowest contrastive loss on validation sets.

**CLIP Baselines.** We compare various CLIP models across unconfounded datasets for two modalities, including OpenAI-CLIP [64], OpenCLIP [34], PubMedCLIP [18], BioMedCLIP [92], PMC-CLIP[1] and MedCLIP [85]. We evaluate these models in both zero-shot and linear probe scenarios. In zero-shot, GPT-4 generates prompts for each class, and we use the ensemble of cosine similarities between the image and prompts as the score for each class. In linear probing, we use the CLIP models as image encoders to extract features for logistic regression. Additionally, we include DenseNet-121 [32] (fine-tuned on the pretraining datasets with cross-entropy loss) as a baseline for linear probing.

**Results.** Figure 9 shows that our CLIP models perform best in both zero-shot and linear probing scenarios for both modalities. We find that preprocessing the text data with an LLM significantly enhances zero-shot performance. Existing medical CLIP models outperform general CLIP models on X-ray datasets but not on skin lesion images, possibly because X-ray data are more prevalent and accessible in the medical domain. While our CLIP models excel with careful data curation, training converges quickly, suggesting the current contrastive objective might not fully exploit the information from the data, potentially taking shortcuts, such as comparing images from different patients instead of focusing on diseases. Future research should explore more suitable objectives and larger-scale data collections to develop more robust medical foundation models.

## B.2 Baselines

This section outlines the implementation details of all baselines compared with our knowledge bottlenecks for medical image classification. All baselines are run on a single RTX A6000 GPU.

- **ViT-L/14**: We utilize the visual encoders from the CLIP models we pretrained in Sec B.1 and add a classification head for downstream classification datasets. We unfreeze the ViT-L/14 backbone and train all parameters with a learning rate of $1e^{-6}$ and a batch size of 64 for 20 epochs.
- **DenseNet-121**: Similarly to ViT-L/14, we add a classification head to the pretrained DenseNet and train the entire network end-to-end. For X-rays, we use the DenseNet pretrained on MIMIC-CXR by TorchXRayVision [13]. For skin lesion images, we pre-train the DenseNet from scratch on ISIC using a cross-entropy loss. When fine-tuned for downstream classification datasets, we train the DenseNet with a learning rate of $1e^{-5}$, a batch size of 64, and also for 20 epochs.
- **Linear Probe**: We employ visual encoders (ViT-L/14) from pretrained CLIP models to extract features for images in downstream classification datasets. We train a linear layer to map these features into labels for 200 epochs with a learning rate of $1e^{-3}$ and a batch size of 64.
- **LSL** [56]: We fine-tune the pretrained CLIP with contrastive loss on annotated concept data (PubMed bottleneck), using the same data as for concept grounding functions (4.3). Training instances are triplets $(I, c, y)$, where $I$ is the image, $c$ is a textual concept, and $y \in \{0, 1\}$ is a binary label indicating whether the image contains this concept. Given the visual encoder $\mathcal{V}$ and textual encoder $\mathcal{T}$ of the CLIP, the cosine similarity between an image and a concept is $s(I, c) = \cos\left(\mathcal{V}(I), \mathcal{T}(c)\right)$. The contrastive loss function is defined as $\mathcal{L}_{\text{contrast}} = y \cdot \max\left(0, m - s(I, c)\right) + (1 - y) \cdot s(I, c)$, where $m = 0.6$ is the margin. We fine-tune the CLIP with concept annotations for 20 epochs with a learning rate of $1e^{-6}$ and a batch size of 64. After obtaining the fine-tuned CLIP, we extract features and train a linear probe in the same manner as the linear probe baseline.
- **PCBM-h** [91] and **LaBo** [90]: We use their codebases to implement these baselines. The concept alignment in both models is achieved using CLIP to compute the dot product between image and concept features. PCBM-h uses the same bottleneck as KnoBo, which is generated from PubMed. For LaBo, we employ GPT-4 to generate candidate concepts for submodular selection.

## B.3 KnoBo Details

This section provides additional details about the implementation of KnoBo.

**Medical Corpus.** We utilize a comprehensive medical corpus for retrieval-augmented generation, detailed as follows: (1) PubMed (5.5M docs, 156.9M snippets); (2) StatPearls (9.3K docs, 301.2K

---

[1]https://huggingface.co/ryanyip7777/pmc_vit_l_14

You are an experienced radiologist [dermatologist]. You are summarizing knowledge about '**QUERY**' from chest X-rays [skin lesion images]. Here are the documents retrieved from the corpus:

**RETRIEVED_DOCUMENTS**

I want you to filter and summarize the information in these documents and generate knowledge in the form of *binary questions*, e.g., "Is there lung opacity?" ["Is the lesion asymmetric?"].
Please follow instructions below strictly:
1. Those knowledge will be used to guide the diagnosis on chest X-rays [skin lesion images], so they must be *visually identifiable* from chest X-ray [skin lesion images] only.
2. The binary questions should be concise and not too specific which can be reused for different cases.
3. The binary questions must not contain the class(disease) name, e.g., you *must not* generate "Is there cardiomegaly?" ["Is the lesion malignant?"] as the knowledge for "Cardiomegaly" ["Malignant Lesion"].
4. If there is not much information in the some documents, you can ignore those documents. If none of the documents contain useful information, you can skip this task by typing 'skip'.
5. Answer with the following format: question | document ID | reference sentence, e.g., Is there lung opacity? | 1234 | lung opacity is a common finding for ... [Is the lesion asymmetric? | 1234 | asymmetric lesion is a common finding for ...]

Please answer without additional information and do not add numbers or bullet points in the answer.

Figure 5: Prompt template for retrieval-augmented concept bottleneck generation. The text in the square brackets is words that need to be changed when using this prompt for skin lesion images.

| LLM | Chest X-ray Datasets | | | | | | Skin Lesion Datasets | | | | | |
|---|---|---|---|---|---|---|---|---|---|---|---|---|
| | ID | OOD | $\Delta \downarrow$ | Avg | Unconfound | Overall | ID | OOD | $\Delta \downarrow$ | Avg | Unconfound | Overall |
| Flan-T5 | 89.7 | **58.8** | **30.9** | **74.3** | **73.1** | **73.7** | 86.0 | 70.5 | 14.1 | 78.3 | 78.1 | 78.2 |
| GPT-4 | **89.9** | 56.8 | 33.1 | 73.3 | 72.9 | 73.1 | 86.0 | **71.6** | **14.4** | **78.8** | **78.5** | **78.6** |

Table 10: Comparison of using different LLM annotating concepts on clinical reports.

snippets); (3) Textbooks (18 docs, 125.8K snippets); (4) Wikipedia (6.5M docs, 29.9M snippets). The StatPearls, Textbooks, and Wikipedia sources are obtained from MEDRAG [88]. Unlike the abstract-only approach of PubMed in MEDRAG, we utilize full articles from PubMed, including all paragraphs. We employ the retrieval codebase of MEDRAG and select BM25 as the ranking function.

**Retrieval-augmented Concept Bottleneck Generation.** Figure 5 illustrates the prompt template we use to generate concepts from documents. We retrieve the top 10 documents for each query as context for the large language model (GPT-4) to generate concepts. After generating concepts, we validate each concept based on three criteria before inclusion in the bottleneck: (1) the concept must be distinct from existing concepts; (2) it must be visually identifiable from the image; and (3) there must be sufficient positive and negative instances in the pretraining corpus to support training its grounding function. A concept is added to the bottleneck only if it meets all three criteria, as judged by another language model (GPT-4). We initially target 200 concepts per bottleneck but ultimately select the top 150 with the highest grounding accuracy for inclusion. This selection is due to some concepts lacking sufficient reports to effectively train their grounding functions, making 150 the minimum size for all the bottlenecks we construct. Table 6 shows examples of generated concepts.

**Concept Grounding.** We use a language model to annotate 2,000 clinical reports for each concept from the pretraining corpus. To efficiently label reports and achieve a balance of positive and negative examples, we retrieve the top 1,000 reports showing high textual similarity (measured by Sentence Transformer [65]) to the concept as its potential positive examples and randomly sample another 1,000 for potential negatives. We use Flan-T5-XXL [10] as the underlying large language model. Specifically, the annotation task is treated as a next token prediction, similar to the approach proposed by McInerney et al. [54], where we compare the probabilities of the next token being Yes or No to determine if the report contains the concept. Table 10 shows no big difference in final classification performance when using Flan-T5 versus GPT-4 for annotating concepts on reports.

# C  Additional Analysis

This section presents additional analysis and ablation studies on our method.

## C.1  Details about Deep Image Priors

We provide further details about the deep image prior experiments in Sec 3.

| Feature | Natural Image Datasets | | | | | |
|---|---|---|---|---|---|---|
| | **CIFAR-10** | **STL-10** | **ImageNet-10** | **Food-101** | **Flower-102** | **Average** |
| Random | 10.0 | 10.0 | 10.0 | 1.0 | 1.0 | 6.4 |
| Pixel Value | 21.2 | 24.7 | 22.4 | 3.0 | 8.5 | 16.0 |
| ConvNext-L$^*$ | 25.6 | 29.1 | 32.6 | 3.7 | 11.3 | 20.5 |
| ViT-L/14$^*$ | **33.3** | **40.6** | **47.2** | **8.9** | **23.6** | **30.7** |

| Feature | X-ray Datasets | | | | | |
|---|---|---|---|---|---|---|
| | **Pneumonia** | **COVID-QU** | **NIH-CXR** | **Open-i** | **VinDr-CXR** | **Average** |
| Random | 50.0 | 25.0 | 16.7 | 33.0 | 14.3 | 27.8 |
| Pixel Value | **77.6** | 66.9 | **43.1** | **58.3** | **20.0** | **53.2** |
| ConvNext-L$^*$ | 62.5 | 59.2 | 38.9 | 57.5 | **20.0** | 47.6 |
| ViT-L/14$^*$ | 67.6 | **68.0** | 40.4 | 57.2 | **20.0** | 50.6 |

| Feature | Skin Lesion Datasets | | | | | |
|---|---|---|---|---|---|---|
| | **HAM10000** | **BCN20000** | **PAD-UFS-20** | **Melanoma** | **UWaterloo** | **Average** |
| Random | 14.3 | 25.0 | 50.0 | 50.0 | 50.0 | 37.9 |
| Pixel Value | 65.9 | 39.4 | **58.0** | 74.5 | **70.0** | **61.6** |
| ConvNext-L$^*$ | 66.9 | 37.1 | 53.5 | 68.3 | 50.0 | 55.8 |
| ViT-L/14$^*$ | **67.3** | **45.9** | 54.0 | **84.8** | 50.0 | 61.5 |

Table 11: Linear Probe results of different features on five natural image datasets, five X-ray datasets, and five skin lesion datasets. $^*$ denotes the network is randomly initialized without any training.

**Datasets.** We evaluate the deep image prior on three categories of images. The X-ray and skin lesion images are the same as those described in the unconfounded datasets section (Sec A). For natural images, we select five datasets: (1) CIFAR-10 [43], (2) STL-10 [11], (3) ImageNet-10 [70][2], (4) Food-101 [7], and (5) Flower-102 [59].

**Setup.** We employ the vision backbones ViT-L/14 [17] and ConvNext-L [51], both implemented by OpenCLIP [34], and initialize them using Kaiming initialization [28] following PyTorch's default settings[3]. Both backbones extract feature vectors of size 768. For the pixel value baseline, we convert the image to grayscale, resize it to $28 \times 28$, and then flatten it into a vector of 784 dimensions. We use the first 768 values of this pixel vector to match the size of the deep features. All features are passed through a linear layer to predict the labels with a learning rate of $1e^{-3}$, a batch size of 64 for 200 epochs. Additionally, we include a random baseline for comparison.

**Results.** Table 11 displays the full results of all methods across the three image categories. ViT-L/14 excels on natural datasets with significant gains over pixel baselines. ConvNext-L performs worse than ViT-L/14 but is still notably more effective than pixel-based methods. For X-ray images, the pixel baseline clearly surpasses the two networks on almost all datasets, indicating that deep models lack priors or even have harmful priors for X-ray imaging. For skin lesion images, which are closer to natural images, the pixel value baseline performs comparably to ViT-L/14 and better than ConvNext-L. Overall, the results suggest that deep networks lack sufficient priors for medical domains, potentially affecting their generalizability.

### C.2 Full Results on Unconfounded Datasets

Table 12 shows the comprehensive results of all baselines across the 10 unconfounded medical datasets. KnoBo performs competitively in the X-ray category, securing top-1 positions for two datasets and achieving an average ranking of third among all methods. For skin lesion datasets, KnoBo's performance is limited by the smaller scale and lower quality of the pretraining corpus, which is annotated by GPT-4V rather than by human experts. This affects the effectiveness of the grounding functions for lesion concepts. With access to larger-scale and higher-quality pretraining data, KnoBo could potentially close the performance gap with black-box baselines.

---

[2]We use the 10 classes selected by Imagenette: https://github.com/fastai/imagenette.
[3]https://pytorch.org/docs/stable/nn.init.html

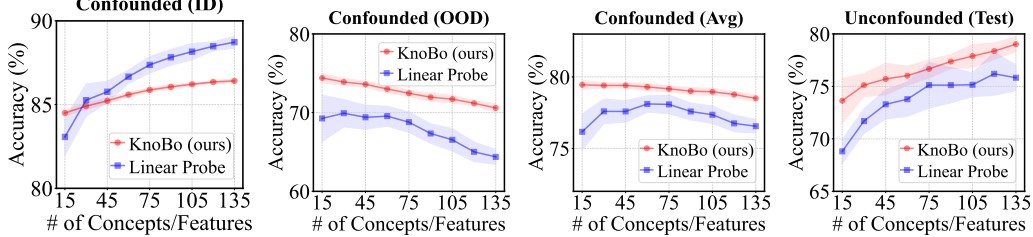

Figure 6: Ablation of **bottleneck sizes** on Skin Lesion datasets. The x-axis is the number of randomly selected concepts (KnoBo) or visual features (Linear Probe). We report the ID, OOD, and domain-average performance on confounded datasets and test the accuracy of the unconfounded datasets.

| Method | Chest X-ray Standard Datasets | | | | | |
|---|---|---|---|---|---|---|
| | **Pneumonia** | **COVID-QU** | **NIH-CXR** | **Open-i** | **VinDr-CXR** | **Average** |
| ViT-L/14 | 84.6 | **96.5** | 66.0 | 67.0 | 37.1 | 70.2 |
| DenseNet | 85.1 | 92.4 | 56.9 | 63.5 | 32.0 | 66.0 |
| Linear Probe | 88.6 | 94.9 | 68.4 | 73.3 | 44.0 | 73.8 |
| LSL | 87.0 | 86.9 | 58.4 | 64.8 | 37.7 | 67.0 |
| PCBM-h | 87.7 | 94.9 | **68.6** | 73.2 | **49.1** | **74.7** |
| LaBo | 88.3 | 91.8 | 66.8 | 68.9 | 44.6 | 72.1 |
| KnoBo (ours) | **90.1** | 88.0 | 66.5 | **73.5** | 47.4 | 73.1 |

| Method | Skin Lesion Standard Datasets | | | | | |
|---|---|---|---|---|---|---|
| | **HAM10000** | **BCN20000** | **PAD-UFS-20** | **Melanoma** | **UWaterloo** | **Average** |
| ViT-L/14 | **87.1** | **76.6** | **88.5** | **94.1** | 75.0 | **84.3** |
| DenseNet | 79.0 | 69.6 | 69.5 | 91.9 | 45.0 | 71.0 |
| Linear Probe | 82.9 | 71.0 | 86.5 | 93.5 | **80.0** | 82.8 |
| LSL | 81.5 | 67.5 | 84.5 | 92.5 | 60.0 | 77.2 |
| PCBM-h | 82.9 | 70.9 | 86.0 | 93.6 | 75.0 | 81.7 |
| LaBo | 80.6 | 68.5 | 82.5 | 93.6 | 75.0 | 80.0 |
| KnoBo (ours) | 78.2 | 65.6 | 80.0 | 91.5 | 75.0 | 78.1 |

Table 12: Test accuracy on 10 **unconfounded datasets** of two modalities.

## C.3 Ablate Bottleneck Size for Skin Lesion

Figure 6 compares the performance of KnoBo and a Linear Probe across different feature sizes on skin lesion datasets. Mirroring the trends observed in X-rays shown in Figure 4, our interpretable bottleneck representations consistently outperform black-box visual features.

## C.4 Data Efficiency for Concept Grounding

The concept grounding module of KnoBo requires image-report pairs from a multimodal medical dataset. By default, we select 2,000 pairs for training the grounding function for each concept. However, this number can be largely reduced. Table 13 shows KnoBo's performance doesn't decrease too much with fewer training examples. With only 100 reports, KnoBo can obtain 72.1 overall accuracy, compared to 73.7, of which 2,000 reports are used. It highlights that KnoBo achieves much of its performance with a smaller number of examples, suggesting that it is not data-hungry and has the potential to be applied to rare modalities.

Table 13: Ablate the number of clinical reports for training each concept grounding function.

| Reports / Concept | Confd | Unconfd | Overall |
|---|---|---|---|
| 100 | 72.5 | 71.7 | 72.1 |
| 250 | 73.7 | 72.1 | 72.9 |
| 500 | 73.1 | 73.0 | 73.1 |
| 1000 | 73.1 | 72.8 | 73.0 |
| 1500 | 73.5 | 73.2 | 73.4 |
| 2000 | 74.3 | 73.1 | 73.7 |

| Query | | CLIP | Ours (w/ concept grounding) |
|---|---|---|---|
| *Are lung fields clear on both sides?* | Top-3 | | |
| *Is there a visible enlargement of the heart?* | Top-3 | | |
| *Does the lesion have a regularly shaped border?* | Top-3 | | |
| *Does the lesion show dry scaly areas?* | Top-3 | | |

Figure 7: Compare CLIP and our concept grounding function in retrieving images based on a text query. Images marked with green checkmarks are correct retrievals as assessed by medical students.

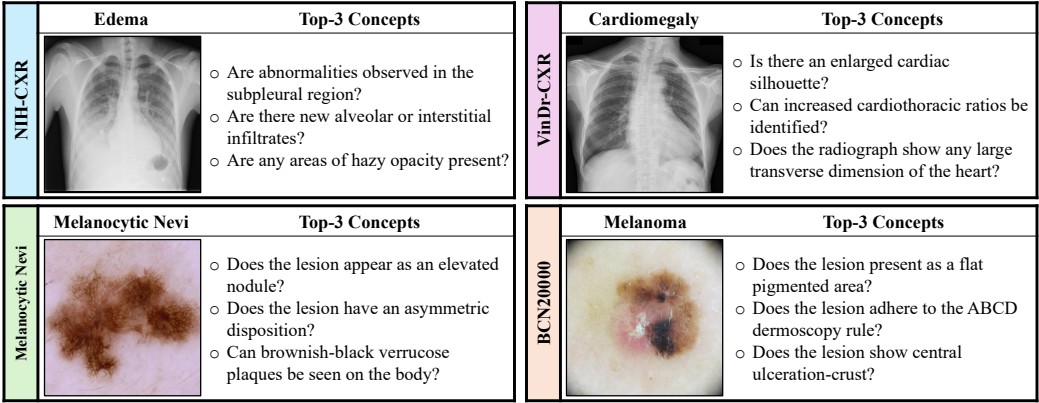

Figure 8: Top-3 concepts for a class ranked by their weights in the linear layer of KnoBo.

# D  Qualitative Examples

Figure 7 compares the CLIP dot product and our concept grounding functions for retrieving images based on text queries. Our method demonstrates superior recall of correct examples compared to CLIP alignments. Figure 8 showcases the top concepts selected based on the linear weights learned by KnoBo, which are most correlated with the corresponding disease class. These examples highlight that the top concepts utilized by KnoBo are essential features doctors use for diagnosing targeted diseases. Table 14 displays examples of failed concepts identified by medical students during human evaluation. Each concept can be traced back to its source document to verify its accuracy. Figure 9 presents examples of GPT-4V-annotated clinical reports for skin lesion images. Notably, the second example demonstrates GPT-4V's capability to assess lesion size using the scale provided in the image.

| Modality | Irrelevant Concepts | Ungroundable Concepts | Repetitive Concepts |
|---|---|---|---|
| X-ray | • Are there any subcutaneous emphysema? 
 • Can you spot elevated hems in the diaphragm? | • Are there changes in appearance that develop slowly? 
 • Can worsening consolidation be observed? | • Is there evidence of ground-glass opacity? 
 • Is there ground glass opacity in both lung fields? |
| Skin | • Is the lesion located on the surface of extremities? 
 • Does it have scales near the lesion? 
 • Are skin lesions involving the face and scalp? | • Is the skin around the lesion rough without pain? 
 • Is the lesion larger than 6mm in diameter? 
 • Is there any change in the size of the lesion over time? | • Does the lesion have regular margins? 
 • Does the lesion possess an irregular shape? 
 • Does the lesion have irregular borders? |

Table 14: We let medical students annotate 300 concepts and categorize the failure cases into three categories: (1) **Irrelevant**: concepts are not essential to diagnosis; (2) **Ungroundable**: concepts cannot be grounded on only one image and (3) **Repetitive**: the concepts are repeated in the bottleneck.

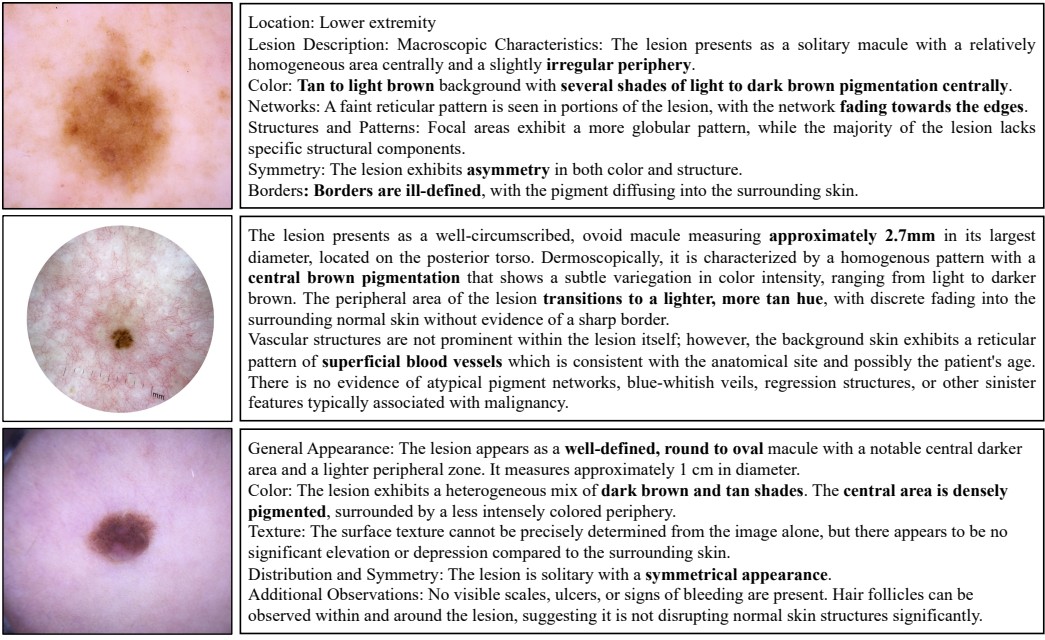

Figure 9: Examples of clinical reports on skin lesion images generated by GPT-4V [61].

