# OpenReview forum: "A Textbook Remedy for Domain Shifts: Knowledge Priors for Medical Image Analysis"
_NeurIPS.cc/2024/Conference — NeurIPS 2024 spotlight_

### Official Review · Reviewer_tMbd · 2024-06-22

**Soundness:** 3
**Presentation:** 3
**Contribution:** 4
**Rating:** 6
**Confidence:** 3

**Summary:**

This paper addresses the issue of deep networks' sensitivity to domain shifts in medical image analysis, particularly for chest X-rays and skin lesion images. The authors propose Knowledge-enhanced Bottlenecks (KnoBo), a concept bottleneck model that incorporates explicit medical knowledge from resources like textbooks and PubMed to improve generalization. KnoBo leverages retrieval-augmented language models to create a clinically relevant concept space and demonstrates a significant performance improvement—32.4% on average—over fine-tuned models across 20 datasets, highlighting PubMed as a particularly effective knowledge resource for mitigating domain shifts.

**Strengths:**

Innovative Integration of Medical Knowledge:
The paper introduces Knowledge-enhanced Bottlenecks (KnoBo), a novel approach that integrates explicit medical knowledge into deep networks. By using retrieval-augmented language models to incorporate clinically relevant factors from medical textbooks and PubMed, the model gains a robust prior that significantly enhances its generalization capabilities.
Comprehensive Evaluation Across Diverse Domain Shifts:
The authors conduct a thorough evaluation of KnoBo across 20 datasets and two imaging modalities (chest X-rays and skin lesion images). This extensive testing demonstrates the model's ability to handle various domain shifts, such as differences in data from different hospitals and demographic confounders, showcasing its practical applicability and robustness.
Significant Performance Improvement:
KnoBo shows substantial performance gains, outperforming fine-tuned models by an average of 32.4% on confounded datasets. This impressive improvement highlights the effectiveness of incorporating medical knowledge for reducing sensitivity to domain shifts, with evaluations indicating that PubMed is a particularly valuable resource for enhancing model performance and information diversity.

**Weaknesses:**

Lack of Application to 3D Medical Imaging:
Issue: The paper primarily focuses on 2D medical imaging modalities, such as chest X-rays and skin lesion images, without addressing the broader and more widely used domain of 3D medical imaging.
Impact: This limitation raises concerns about the generalizability and applicability of the proposed KnoBo model to 3D imaging scenarios, such as CT scans or MRI, which are critical in many clinical contexts. Including evaluations on 3D medical images would significantly enhance the relevance and impact of the research.
Insufficient Comparative Analysis with 3D Domain Adaptation Methods:
Issue: There is a lack of comparative analysis with existing domain adaptation and generalization methods specifically applied to 3D medical imaging.
Impact: The absence of such comparisons makes it difficult to gauge the relative effectiveness of KnoBo in a comprehensive manner, particularly given the significant advancements in 3D domain adaptation techniques. Addressing this gap would provide a more complete evaluation of KnoBo's capabilities and limitations.

**Questions:**

Application to 3D Medical Imaging:
Question: Have you considered extending the KnoBo framework to 3D medical imaging modalities such as CT scans or MRIs? If so, what challenges do you anticipate, and how might they be addressed?
Suggestion: Expanding your research to include 3D medical images would significantly increase the applicability and impact of your work. Discussing potential adaptations and preliminary results for 3D imaging could provide valuable insights and demonstrate the versatility of KnoBo.
Comparative Analysis with State-of-the-Art Methods:
Question: How does KnoBo compare with recent state-of-the-art domain adaptation and generalization methods, especially those tailored for 3D medical imaging?
Suggestion: Including a detailed comparative analysis with advanced domain adaptation techniques, particularly those used in 3D imaging, would strengthen your paper. This would provide a clearer benchmark and highlight the unique advantages and limitations of KnoBo.

---

> ### Author Rebuttal · Authors · 2024-08-06
>
> Thank you for your positive feedback on our work! Here, we address your concern about applications in 3D images.
>
> We agree with you that 3D modalities are important. However, we scoped our study to 2D modalities as these are cheaper in practice. Part of the motivation is to transfer to less-resourced hospitals where X-rays are generally available but more expensive imaging modalities are not (i.e., in developing countries). That being said, we are very concerned with establishing the broad applicability of our work, and so we evaluated 20 datasets within the modalities we considered.
>
> The KnoBo framework is flexible enough to adapt to 3D modalities because the key contribution is the factorization. Many components of KnoBo can be readily reused, such as the concept generation from medical documents. We only need an appropriate 3D encoder (e.g., 3D CNN) and a pretraining dataset. Fortunately, large-scale public 3D medical datasets are available, e.g., DeepLesion [1] has 32,000 annotated CT scans, CT-RATE [2] has 50,188 CT volumes with associated clinical reports, and the recently released CLIP-CT [2] could serve as a backbone.  We agree that 3D would enrich the work, but we feel we have provided significant evidence already that KnoBo is effective and that there is a clear path on how one could apply it to 3D domains in the future.
>
> [1] Yan et al. DeepLesion: Automated Deep Mining, Categorization and Detection of Significant Radiology Image Findings using Large-Scale Clinical Lesion Annotations. 2017.
>
> [2] Hamamci et al. A foundation model utilizing chest CT volumes and radiology reports for supervised-level zero-shot detection of abnormalities. 2024.

---

> ### Author Response · Authors · 2024-08-10
> **Has the rebuttal addressed your concerns?**
>
> Hi,
>
> Do you feel we have provided enough detail about possible extensions of our work to 3D? We will add some of this discussion to the limitations and future work section given extra space. Would you consider updating your rating or are there other weaknesses that we can discuss now during the discussion period?
>
> Thanks!

---

> > ### Comment · Reviewer_tMbd · 2024-08-12
> >
> > Thank you for your response. I would keep my score.

---

### Official Review · Reviewer_f9Nf · 2024-07-10

**Soundness:** 2
**Presentation:** 2
**Contribution:** 3
**Rating:** 6
**Confidence:** 5

**Summary:**

The authors noticed the domain-shift issue of current medical dataset, and after taking inspiration from medical training, they propose giving deep networks a prior grounded in explicit medical knowledge communicated in natural language. The proposed network can incorporate medical knowledge priors to help model make decisions. Basically, the authors assumed that “medical concepts” are robust enough to against the domain shift. Valid assumption, interesting work.

**Strengths:**

The authors noticed the domain-shift issue of current medical dataset, and after taking inspiration from medical training, they propose giving deep networks a prior grounded in explicit medical knowledge communicated in natural language. The proposed network can incorporate medical knowledge priors to help model make decisions. Basically, the authors assumed that “medical concepts” are robust enough to against the domain shift. Valid assumption, interesting work.

**Weaknesses:**

It is very HARD to read and interpret. Please raise examples for better reading experience. Maybe I have some questions after the clearance. Other points could be seeen under the Questions section.

**Questions:**

1. “In 5 such constructed confounds per modality, covering scenarios of race, sex, age, scan position, and hospital, we find models unable to generalize well, dropping over 63% on average over an in-distribution (ID) evaluation.” What are the tasks? Please specify.
	2. “As a baseline, we extract a subset of d pixels directly from the image as the feature without any model-based priors, represented as xp ∈ Rd. We compare the classification performance using x versus xp to probe the efficacy of the vision backbone’s priors.” I didn’t get the point. Why did you want to compare the effectiveness of x and x_p?
	3. It is very hard to read the 4.2 section. I cannot fully understand the algorithm 1. Please raise examples for better reading experience.
	4. Table 1, why the KnoBo always has not the best, or sometimes the worst performance in in-domain dataset?
	5. Is the Structure Prior learnt? Or the questions like “Is there ground-glass opacity?” were pre-defined? How ist he connection between “Is there ground-glass opacity?” and the top-right red box in Figure 1?
	6. Is the training end-to-end? Can you please provide an example for 1) how to pretrain using medical books 2) and given an X-ray image, how the workflow is to get the last result? I hold positive thought towards the study, but it is really hard to understand the workflow.
	7. Is the parameter prior a part of the bottleneck predictor? In figure 1 it seems like parallel, but in the main text it seems like existing sequentially. Hard to understand.
	8. How to determine the best number of concepts?
	9. How did you determine the preferred correlation between the label y and concept c?

---

> ### Author Rebuttal · Authors · 2024-08-06
>
> Thanks for the feedback! We will try to include more examples in future versions. We hope the answers below clarify:
>
> **Q1. What are the tasks?**
>
> We studied the medical image classification tasks in a confounded setting (the medical class names are found in Tables 5 and 6). As explained in the second paragraph of the introduction (Lines 30-34), the samples are confounded with different factors during training/validation (ID) and testing (OOD) time. For example, in an X-ray classification task to classify COVID and normal patients, if we confound the data with sex, the dataset is constructed as follows:
>
> Train/Validation (ID): (COVID, male) (Normal, female)
>
> Test (OOD): (COVID, female) (Normal, male)
>
> A robust model should predict based on COVID-related features rather than spurious correlations (sex in this example).
>
>
> **Q2. Why compare the effectiveness of $x$ and $x_p$?**
>
> This experiment aims to validate our motivation that deep neural networks lack priors for medical domains. $x$ is the feature extracted by a deep network at initialization (i.e., a ViT). $x_p$ is a feature vector constructed by extracting pixel values in the image. If a network has good priors for a domain, we expect the network output at initialization ($x$) to be a more useful representation of the image content than the pixels themselves ($x_p$). For natural images, this is known from prior work on Deep Image Priors, but, as we show in Figure 2, this is not true for medical images. Without good priors, models can easily adopt incorrect hypotheses and rely on spurious features. This leads to catastrophic failures when the spurious features are unavailable in out-of-domain tests and motivates the need to incorporate priors from elsewhere.
>
> **Q3. explain algorithm 1**
>
> Algorithm 1 aims to collect a set of concepts from medical documents. Given the class name (e.g., COVID), we want to extract useful knowledge from medical documents as concepts (e.g., ground glass opacity, a highly indicative feature of COVID). The whole process is a retrieval-augmented-generation task. We first use class names as queries (e.g., COVID) to retrieve relevant documents from a corpus (e.g., PubMed). The retrieved document may contain useful information, e.g., in the upper left of Figure 3, the document says, “the most frequent radiological patterns found were ground-glass opacities.” We feed these documents as contexts for an LLM, which extracts the information as a concept (ground glass opacity). We repeat this process using generated concepts as queries until a predefined number of concepts is reached.
>
> **Q4. Why doesn’t KnoBo always have the best ID performance?**
>
> This is because, in the confounded settings (explained in Q1), models can achieve high ID performance by taking shortcuts using spurious correlations, e.g., learning to classify males and females instead of focusing on diseases. Models that achieve high ID performance in this way will have catastrophic failures on the OOD test (see OOD numbers of ViT-L/14 and DenseNet in Table 1). In this case, a robust model must have a high average performance across ID and OOD. There is a natural tradeoff between ID and OOD, but our models strike the best balance.
>
> **Q5. Is the Structure Prior learnt? The connection between “Is there ground-glass opacity?” and the top-right red box.**
>
> As explained in Q3, the structure prior is not learned from training instances but instead constructed from medical documents.
>
> The top-right red box illustrates the concept grounding module (Sec 4.3, Lines 180-188). Given a concept (e.g., is there ground-glass opacity?) and an X-ray image, we aim to estimate the probability of this concept existing on this X-ray.
>
> To achieve this goal, We train a binary classifier for each concept. We obtain the positive and negative examples for each concept by leveraging a pretraining dataset with (X-ray, clinical report) pairs. We can estimate the presence of a concept in each example by prompting an LLM to generate a response indicating whether the clinical report implies the concept. For example, if the clinical report mentioned related terms about “ground-glass opacity,” it is a positive example of this concept. Otherwise, it will be a negative example. With positive and negative examples for a concept, we can train a binary classifier to predict the concept given an X-ray image.
>
> **Q6. Is the training end-to-end?**
>
> No. To be computationally feasible, the training has stages: (1) constructing the prior from documents (explained in Q3), (2) concept grounding (explained in Q5), and (3) linear layer for final label prediction. Given an X-ray image, the workflow is: (1) execute the concept grounding functions to get the probabilities on all concepts in the structure prior, (2) use the concept probabilities as the input to a linear layer to get the final label prediction, e.g., predict COVID or Normal. Please also refer to Lines 142-144 for Preliminary on Concept Bottleneck Model and Sec 4.3 for details about the workflow.
>
> **Q7. Is the parameter prior a part of the bottleneck predictor?**
>
> No. As explained in Sec 4.4, the parameter prior is used to regularize the learning of the linear layer. We will compute the L1 distance between the linear layer and the parameter prior as the loss to align the learned parameters with priors.
>
> **Q8. How to determine the best number of concepts?**
>
> We did not tune this hyperparameter much and used 150 concepts. In Figure 4, we examine the impact of the number of concepts on ID and OOD performance. More concepts lead to better ID but worse OOD so that an appropriate trade-off can be selected for a problem.
>
> **Q9. How did you determine the preferred correlation between the label y and concept c?**
>
> A language model (explained on Line 198) determines the preferred sign of the correlation between y and c.
>
> Please do not hesitate to ask more questions, and we will be happy to answer them. We look forward to your response!

---

> ### Author Response · Authors · 2024-08-10
> **Has the rebuttal clarified sufficiently?**
>
> Hi,
>
> Has our rebuttal clarified sufficiently that you feel you can judge our work? Are there other aspects that we can clarify for you to make an assessment now that we have provided answers to your question?
>
> Thanks!

---

> > ### Comment · Reviewer_f9Nf · 2024-08-10
> >
> > Thank you for resolving all my questions. Now I have no more questions. I have adjusted my ratings. Nice work!

---

### Official Review · Reviewer_T8n7 · 2024-07-13

**Soundness:** 3
**Presentation:** 4
**Contribution:** 3
**Rating:** 7
**Confidence:** 3

**Summary:**

The presented paper addresses the challenge of domain shifts in medical image classification, where conventional neural networks often lack effective priors for medical datasets. The authors introduce KnoBo, a novel class of concept bottleneck networks that integrate medical knowledge priors to enhance neural network performance in medical image classification tasks. KnoBo has three primary components: 1) Structure Prior:  leverages medical documents to construct a knowledge bottleneck; 2) Bottleneck Predictor:  maps images onto concepts; 3) Parameter Prior: imposes constraints on learning the linear layer's parameters, ensuring that the network's predictions remain consistent with the established medical knowledge. The proposed method of KnoBo is evaluated using two medical datasets, ISIC and X-ray, under both confounded and unconfounded settings.

**Strengths:**

The paper is well-organized and easy to follow. The authors propose a novel approach to incorporating medical knowledge priors into existing neural networks by optimizing three distinct terms. The theory behind each term is solid and clearly explained, making the methodology easy to understand. One notable aspect is the parameter prior (Sec. 4.4), which adds interpretability by aligning the estimations with medical concepts. They also use human evaluation on the learned concepts.

The authors conduct a comprehensive set of experiments to evaluate their proposed method, covering several scenarios, including confounded and unconfounded data. This broad range of experiments supports the validity of their findings.

The ablation studies are thorough, encompassing five knowledge sources, appropriate baselines, and various bottleneck sizes. These ablations provide valuable insights into the proposed method's performance and sensitivity.

**Weaknesses:**

The paper's primary weakness is its reliance on content generated by LLMs to create new medical concepts. Although GPT-4 is one of the most advanced models for generating concepts, it can still produce hallucinations, especially when the generated concepts are loosely aligned with common terms such as hair, skin color/tone, gel bubble, and rules (in the context of skin lesions).

**Questions:**

- Whats is the main reason to define $ \Delta = | ID - OOD |$ as robustness measure (row 257)? In my opinion, this metric is more related to some "fair predictions" concept, since it measures how ID and OOD performances differ.
- Do the authors know why the diversity values for skin lesion datasets (Table 3, last column) are much lower than those for X-rays?

**Limitations:**

Not a limitation, but I recommend incorporating a filtering scheme into the concept generation process to enhance the paper. While the authors have conducted human evaluations, relying solely on such assessments may not be feasible due to the limited availability of human resources. Refining the concept generation could be achieved by improving the prompts or employing a medical foundation model to assess the quality of the generated concepts about the target image.

---

> ### Author Rebuttal · Authors · 2024-08-06
>
> Thank you for your positive feedback on our work! Here is our reply to your questions:
>
> **Q1. Hallucinations of GPT-4**
>
> This is a valid concern, but the way we use GPT-4 highly encourages it to directly copy information from documents instead of inventing it. Our concept generation is conditioned on medical documents, which differs from previous work [1] that solely relies on LLM. We employ LLM as a text extraction tool, which basically copy-paste or paraphrases. In addition, each concept is attributed to a document that medical professionals can fact-check. Table 12 of the Appendix and supplementary materials (HTML files in the `example_concepts/` folder) show example concepts with corresponding reference documents.
>
> We examined concepts from the X-ray bottleneck. 80% of the concepts were word-for-word copies of a feature in a sentence that the authors of a document said they examined. 14% included some paraphrases but were still grounded in the document and never completely invented (sometimes a combination of multiple factors in different places in the document). 6% were of confusing origin and potentially hallucinated but still mentioned relevant factors. This aligns with the analysis of the bottlenecks we did with medical students in section C.4 of the appendix.
>
> **Q2. Robustness measure ($\Delta$)**
>
> Our analysis is focused on multiple metrics at once. The delta measure $\Delta$ needs to be combined with the other metrics (ID, OOD) when measuring models’ robustness (it is trivially minimized by creating classifiers with chance performance). Our perspective is that a robust model should have high performance across ID and OOD while maintaining a small drop to demonstrate its robustness to domain shifts.
>
> **Q3. Diversity values for skin lesion datasets are much lower than those for X-rays**
>
> This is because X-ray is a more information-dense modality as it shows multiple organs (e.g., lung, heart, stomach). Skin lesion diagnosis requires fewer medical factors in practice. Moreover, X-ray is the most accessible medical imaging modality, resulting in more available medical documents.
>
> **Q4. Filtering scheme of concept generation**
>
> Thank you for the suggestion! To clarify a bit, we don’t have a phase in our method that requires medical professionals, although we used them later to verify the validity of our approach. We designed three filtering criteria to ensure the concepts are diverse and visually identifiable. (Appendix B.3, Line 712-717) As you suggest, more filtering of the concepts using auxiliary signals would likely substantially improve the results and is something we are exploring.
>
> [1] Yang et al. Language in a bottle: Language model guided concept bottlenecks for interpretable image classification. CVPR 2023.

---

> > ### Comment · Reviewer_T8n7 · 2024-08-10
> >
> > Thank you for your response.
> >
> > The authors have adequately addressed my primary concerns, and I have no further questions. I will maintain my previous rating.

---

> ### Comment · Reviewer_z571 · 2024-08-12
> **Regarding Hallucinations**
>
> I just want to add on here as I believe RAG is quite an effective and reasonable measure to prevent hallucinations by far. Plus, the number/stats they reported in the rebuttal seem pretty convincing.

---

### Official Review · Reviewer_z571 · 2024-07-13

**Soundness:** 3
**Presentation:** 3
**Contribution:** 3
**Rating:** 7
**Confidence:** 3

**Summary:**

This paper proposed Knowledge-enhanced Bottlenecks (KnoBo) to leverage the concept bottleneck model (CBM) to improve model robustness to various domain shifts and confounders. KnoBo explores using retrieval augmented generation to generate concept space with a large number of concepts using a large language model (i.e., GPT-4). KnoBo brings significant overall improvement when large-scale image-text pairs are available for concept grounding. A comprehensive evaluation across various scenarios and various baselines confirms the effectiveness of KnoBo in reducing the gap between in-distribution and out-of-distribution performance.

**Strengths:**

1. This paper explores exploiting LLM and retrieval augmentation generation, which enables building large concept space from the large-scale knowledge base.
2. Overall, the presentation is clear. Technical details are clearly presented in the supplemental material. However, some results are a bit confusing and lack discussion of limitation and failure cases.
3. Evaluation is comprehensive.

**Weaknesses:**

1. The concept grounding seems pretty data-hungry, requiring large-scale multi-modal datasets for pretraining (e.g., MIMIC-CXR with over 300k image-text pairs). For smaller datasets (ISIC with about 60k images with generated captions from LLM), the proposed KnoBo only brings marginal overall improvement. This requirement significantly limits the broad application of the proposed framework in the data-scarcity domain, such as cancer/tumor classification.
2. Lack of a detailed discussion of limitations (one one-line limitation in Section 7) and presentation of failure cases. No failure cases (wrong classification / wrong concept) were included in either the main text or supplemental material.
3. Some results are confusing to the reviewer (See No. 2 in the Questions section).

**Questions:**

1. In Table 1, the best performance of ISIC-age should be ViT-L/14 and LSL, not the proposed KnoBo (which is bolded).
2. The KnoBo with 150 concepts has an average OOD Acc of 58.8 across Chest X-ray Datasets, as stated in Tables 2 and 4. Why does Figure 4 indicate that KnoBo with 135 concepts has an accuracy below 50 for OOD X-ray datasets? As the performance of OOD datasets decreases with the increased number of concepts, these two results seem contradicted.
3. Typo in the title of Algorithm 1 in Section 4.2. Should be 'retrieval'

**Limitations:**

As explained in the Weaknesses and Questions sections.

---

> ### Author Rebuttal · Authors · 2024-08-06
>
> Thank you for your constructive feedback. Here we address your comments:
>
> **Q1. The concept grounding seems pretty data-hungry.**
>
> This is a valid concern, but while we sample from a large dataset like MIMIC, in reality, we only use 22k total examples. Training the concept grounding component is just learning a linear classifier, so we sampled **a small subset of the data**.
>
> On average, each X-ray grounding function uses 1,523 examples, and each skin lesion grounding function uses 1,342 examples. This can be reduced significantly with little loss in performance.
>
> The table below shows the performance of KnoBo on X-ray datasets using varied sizes of samples for training each grounding function. It highlights that KnoBo achieves much of its performance with a small number of examples, suggesting that it is not data-hungry.
>
> | n. of examples | Confounded | Unconfounded | Overall |
> |-------------------------|------------|--------------|---------|
> | 250                                      	| 73.7   	| 72.1     	| 72.9	|
> | 500                                    	| 73.1   	| 73.0     	| 73.1	|
> | 1000                                   	| 73.1   | 72.8     	| 73.0	|
> | 1500                                   	| 73.5   | 73.2     	| 73.4	|
> | 2000                                   	| 74.3   | 73.1     	| 73.7	|
>
> **Q2. For skin lesion datasets KnoBo only brings marginal overall improvement.**
>
> Empirical challenges within the skin lesions datasets are not because of a lack of data but because the datasets have very strong non-generalizable cues (skin colors, hairs, etc). For this reason, it is easy to fine-tune models to high in-domain performance and learn non-transferable predictors.
>
> Our overall metric averages both in-domain performance and out-of-distribution performance. KnoBo is much better out of distribution (see OOD and Avg in Table 2, shows differences of **+22.9** and **+6.7**, respectively), but when averaging in domain performance on datasets with no explicitly identified confounds (unconfd), helps more moderately (+.3). We believe that OOD measures more accurately capture if models are able to learn the correct hypothesis and therefore the more important measure of if a model is trustworthy in real applications.
>
> **Q3. Confusing results in Tables 2 and 4.**
>
> In Figure 4, we deactivated the prior loss for KnoBo, which resulted in lower OOD performance. In this ablation, we focus on comparing different inputs (black-box features vs. concept scores) and their sizes while removing prior loss forms a more comparable setting. We provide the **updated Figure 4 in the PDF of the global response**, which includes the curves of KnoBo with prior loss. We will clarify this experimental setup in the final draft of our paper.
>
> **Q4. Failure cases and limitation section.**
>
> Failure cases are difficult to examine and generally not very accessible to non-medical professionals (in the main text, we tried to avoid adding examples that might make some readers uncomfortable if they are squeamish about medical images). Sections C.4 (Table 11) and Section D (Figures 8 and 9) of the appendix contain some of this, where we recruited medical students to evaluate limitations.
>
> The medical students annotated 300 concepts and often found the information in our bottlenecks highly relevant, but some were not groundable (**see Table A in the PDF for examples**). This did not vary much by information source, but skin lesion information was judged to be easier to ground than X-ray information. In a possible camera-ready version, given the extra page, we can try to integrate some more of this content in the main body of text, including example failures, and expand the limitations section to more explicitly discuss the data efficiency of our methods.
>
> **Q5.Typos.**
>
> We have carefully proofread the manuscript again to correct the issues.

---

> > ### Comment · Reviewer_z571 · 2024-08-12
> > **Response to the rebuttal**
> >
> > I thank the authors for their time and effort in preparing the rebuttal. The rebuttal resolved some of my questions, and I appreciate the explanation and revision for Figure 4 and the supplement of failure concepts in the attachment. I have some remaining questions:
> >
> > 1. About the examples per grounding functions, does this part correspond to lines **177-179**? Could the author point out or supply how many grounding functions are used for each dataset, how they are selected, and how is the # of grounding functions confirmed? I tried but failed to find these details in the manuscript. If you use 22k examples in x-ray, you should have (22k/2k=11) grounding functions? Is that different from the skin dataset? And why does each x-ray grounding function use 1,523 examples when you actually supply 2,000 examples per grounding function?
> >
> > 2. For the failure case, would it be possible to supply some failure examples of KnoBo in Fig. 7 (i.e., examples without the green checkmarks)?
> >
> > Thank you very much, and I am looking forward to your reply. Great work!

---

> > > ### Author Response · Authors · 2024-08-12
> > >
> > > Thanks for the reply! Here, we reorganize your questions and answer them as follows:
> > >
> > > **Q1. About the examples per grounding functions, does this part correspond to lines 177-179?**
> > >
> > > Yes. Sec 4.3 explains the method.
> > >
> > > **Q2. how many grounding functions are used for each dataset, how they are selected, and how is the # of grounding functions confirmed?**
> > >
> > > 150 grounding functions are used for each modality. As explained in lines 717-720, we initially use our retrieve-augmented generation algorithm to generate 200 concepts. However, some concepts may not have enough examples in the corpus to support training the grounding functions. Therefore, we only select the top 150 grounding functions ranked by their concept prediction accuracy on the validation sets.
> > >
> > >  **Q3. If you use 22k examples in x-ray, you should have (22k/2k=11) grounding functions? Is that different from the skin dataset?**
> > >
> > > Each example can be reused to train multiple grounding functions, as each clinical report can contain information on different aspects. For instance, example A can be used as a positive example for concepts 1, 2, and 3 and serve as a negative example for concepts 7, 8, 9, etc. Therefore, considering the overlaps, we use 22k examples in total. The skin lesion dataset uses a similar number of examples.
> > >
> > > **Q4. And why does each x-ray grounding function use 1,523 examples when you actually supply 2,000 examples per grounding function?**
> > >
> > > We set 2000 as the upper bound for examples, but when training each binary grounding function, we want to balance positive and negative examples through subsampling. Therefore, the actual number of training samples is 2$\times$min(positive examples, negative examples), which is smaller than 2000.
> > >
> > > **Q5. For the failure case, would it be possible to supply some failure examples of KnoBo in Fig. 7 (i.e., examples without the green checkmarks)?**
> > >
> > > This is definitely possible, and we will include them in the next version.
> > >
> > > Thanks again for raising those questions! We will clarify those details in the next version of the paper.

---

> > > ### Author Response · Authors · 2024-08-12
> > > **Grounding Function Data Requirements Clarification**
> > >
> > > Apologies, we wanted to clarify the question of data efficiency in the grounding function further (and there was a small typo in the images required before). The mechanism for constructing them is:
> > >
> > > 1. given a concept name (i.e. ground class opacities for chest x-rays), and a pretraining corpus of (image,report) pairs (i.e. MIMIC), reports are sorted by S-BERT embedding similarity to the concept. The list is truncated to a fixed number of the most similar ones (i.e 2k in the manuscript, but as few as 100 below).
> > > 2. An LLM is used to predict if the report positively or negatively affirms that the concept is present in the image, and the output is used as a label.
> > > 3. A linear classifier is trained given a balanced subsample of positive and negative instances for the concept
> > >
> > > The data requirements are that of training a linear classifier on a fixed feature representation and so practically is very small. Furthermore, between the set of all classifiers that need to be trained, there may be correlations between which reports likely mention information relevant to their construction.
> > >
> > > In the paper, we train 150 concept classifiers, so this means that there would be a maximum of 2000 * 150 = 300k image-report pairs required for training.  Practically, the actual number of samples is much lower because of correlations, 50k. Below is an updated table containing the number of samples required to train concept classifiers. With as few as 100 reports /concept, requiring at most 15k samples (although practically 9k), performance decreases very moderately.
> > >
> > > We acknowledge that this level of detail was not clear from the original manuscript and is an important part of how the method works and so will expand it in any future revisions. We will include a small "algorithm" describing the process.
> > >
> > > |reports / concept | maximum samples | actual samples|Confounded|Unconfounded|Overall
> > > | -------- | ------- | -------- | ------- | -------- | ------- |
> > > |100 | 15,000 | 9,060 | 72.5 | 71.7 | 72.1 |
> > > |250	| 37,500 | 10,603 | 73.7|	72.1	|72.9| | |
> > > |500| 75,000 | 19,288 | 73.1|	73.0|	73.1| | |
> > > |1000|150,000 | 33,113 | 73.1|72.8|73.0| | |
> > > |1500|225,000 | 43,229 | 73.5|73.2|73.4| | |
> > > |2000|300,000 | 50,071 | 74.3|73.1|73.7| | |

---

> > > > ### Comment · Reviewer_z571 · 2024-08-13
> > > > **Thanks for the clarification**
> > > >
> > > > Thank you for the reply and clarification. I was confused about the number of examples for grounding functions after receiving the first rebuttal and, therefore, did not update my score. This clarification helps a lot.
> > > >
> > > > Before the end of the discussion and providing my revised score, I have one last question: The title of this manuscript contains a broad term of medical image analysis, while the current version merely focuses on classification. Given the fact that the ISIC dataset also provides a segmentation mask, I wonder if the authors have explored applying KnoBo in the context of segmentation and OOD/confounded generalization.

---

> > > > > ### Author Response · Authors · 2024-08-13
> > > > > **Thanks for interesting discussion! Extension to detection**
> > > > >
> > > > > Thanks for the reply. We appreciate the engagement and feel this discussion is making the work much better.
> > > > >
> > > > > In general, we believe classification is often an important sub-component of medical image analysis pipelines and KnoBo shows how to make that aspect controllable with priors via interpretable design. Segmentation (and related problems like detection and registration) is an important problem, although we have not thoroughly explored it yet. In the case of segmentation though, many model architectures decouple region prediction and classification (i.e. works like Mask R-CNN or Yolo have separate heads for predicting bounding boxes and their classifications) and KnoBo almost directly applies. In terms of making the spatial part of predictions interpretable and having appropriate priors, we may need a different mechanism than concept bottlenecks. It would also be interesting to analyze if confounding errors for segmentation are coming from the spatial or classification components. We can include a discussion of this in the future work / limitations section.

---

> > > > > > ### Comment · Reviewer_z571 · 2024-08-13
> > > > > >
> > > > > > It would be great if you could add a hint of discussion about potentially applying KnoBo to segmentation/detection settings, and I think it would increase the paper's influence. If space not allowed, it could be put in the appendix. I hope the author can manage to include all the committed changes (to my review and others' review) in the final version.
> > > > > >
> > > > > > I feel like the rebuttal and discussion adequately address my questions and concerns. Therefore, I'm happy to raise my rating of presentation from 2 to 3, my rating of contribution from 2 to 3, and my overall rating from 5 to 7. Congrats on the work!

---

> > > > > > > ### Author Response · Authors · 2024-08-13
> > > > > > > **Thanks for all of the feedback and suggestions!**
> > > > > > >
> > > > > > > Thanks for all of the feedback and suggestions!

---

> ### Author Response · Authors · 2024-08-10
> **Has the rebuttal addressed your concerns?**
>
> Hi,
>
> Has our rebuttal addressed the weakness you felt the paper had? In particular, we strongly feel our method is not data-hungry and we have a fairly extensive discussion of problems (currently in the appendix) that could be moved up given extra space. Would you consider updating your rating or are there other weaknesses you feel we can address during this discussion period?

---

### Author Rebuttal · Authors · 2024-08-06

We deeply appreciate the time and effort all reviewers have contributed. We feel very encouraged to see that all the reviewers have overall positive attitudes towards our work. Reviewers found our work interesting (f9Nf) and praised the **novelty of our method** (T8n7, tMbd) with **comprehensive evaluations** (z571, T8n7, tMbd) and **clear presentation** (z571, T8n7, tMbd).

We respond to each reviewer's comments individually and hope your concerns can be addressed. Again, we appreciate your time and expertise in the evaluation process.

---

### Decision · Program_Chairs · 2024-09-25

**Decision:**

Accept (spotlight)

**Comment:**

All reviewers are in agreement and there is a strong consensus regarding the novelties presented in the paper, the thoroughness of the evaluations, and the clarity of the writing. The paper demonstrates significant contributions to the field and provides comprehensive experiments that validate the proposed approach, making it a well-received submission. However, some ethical concerns were raised during the review process, which the ethics reviewers believe are addressable by the authors. These concerns mainly revolve around specific aspects of data usage and consent, but with appropriate revisions and clarifications, the paper is expected to meet ethical standards. The authors are advised to address all raised issues in the final paper.